# GENERALIZED DEMOGRAPHIC PARITY FOR GROUP FAIRNESS

**Zhimeng Jiang**[1], **Xiaotian Han**[1], **Chao Fan**[1], **Fan Yang**[2], **Ali Mostafavi**[1], **and Xia Hu**[2]
[1]Texas A&M University, [2]Rice University

## ABSTRACT

This work aims to generalize demographic parity to continuous sensitive attributes while preserving tractable computation. Current fairness metrics for continuous sensitive attributes largely rely on intractable statistical independence between variables, such as Hirschfeld-Gebelein-Renyi (HGR) and mutual information. Statistical fairness metrics estimation relying on either tractable bounds or neural network approximation, however, are not sufficiently trustful to rank algorithms prediction bias due to lack of estimation accuracy guarantee. To make fairness metrics trustable, we propose *Generalized Demographic Parity* (GDP), a group fairness metric for continuous and discrete attributes. We show the understanding of GDP from the probability perspective and theoretically reveal the connection between GDP regularizer and adversarial debiasing. To estimate GDP, we adopt hard and soft group strategies via the one-hot or the soft group indicator, representing the membership of each sample in different groups of the sensitive attribute. We provably and numerically show that the soft group strategy achieves a faster estimation error convergence rate. Experiments show the better bias mitigation performance of GDP regularizer, compared with adversarial debiasing, for regression and classification tasks in tabular and graph benchmarks [1].

## 1 INTRODUCTION

Fairness problem has attracted increasing attention in many high-stakes applications, such as credit rating, insurance pricing and college admission (Mehrabi et al., 2021; Du et al., 2020; Bellamy et al., 2018), the adopted machine learning models encode and even amplify societal biases toward the group with different sensitive attributes. The majority of existing fairness metrics, such as demographic parity (DP) (Feldman et al., 2015), equal odds (EO) (Hardt et al., 2016), presumably consider discrete sensitive variables such as gender and race. In many real-world applications including urban studies and mobility predictions (Tessum et al., 2021), however, individuals' sensitive attributes are unavailable due to privacy constraints. Instead, only aggregated attributes presenting in continuous distributions are available, and thus fairness requires unbiased prediction over *neighborhood* or *region*-level objects. Additionally, the sensitive attributes, such as age and weight, are inherently continuous (Mary et al., 2019; Grari et al., IJCAI'20). The widely existing continuous sensitive attributes stimulate further fairness metrics definition and bias mitigation methods.

Existing fairness metrics on continuous sensitive attributes rely on the statistical measurement of independence, such as Hirschfeld-Gebelein-Renyi (HGR) maximal correlation coefficient (Mary et al., 2019) and mutual information (Jha et al., 2021; Creager et al., 2019), which are computation-intractable due to the involved functional optimization. Note that the mutual information involves the ratio of probability density function, it is intractable to directly estimate mutual information via probability density function estimation due to the sensitivity over probability density function, especially for the low probability density value. Previous works (Roh et al., 2020; Lowy et al., 2021; Cho et al., 2020) adopting mutual information or variations as regularizer, however, rely on tractable bound, or computationally complex singular value decomposition operation (Mary et al., 2019), or training-needed neural network approximation (Belghazi et al., 2018), such as Donsker-Varadhan representation (Belghazi et al., 2018), variational bounds (Poole et al., 2019). Nevertheless, it is

---

[1]Codes are available at https://github.com/zhimengj0326/GDP

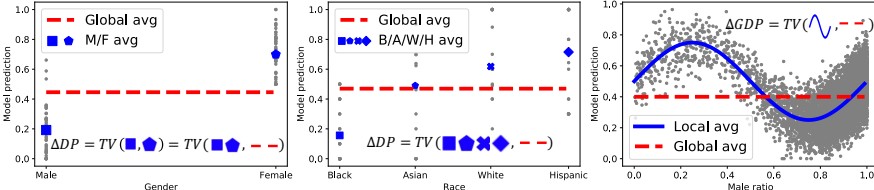

Figure 1: Illustration of demographic parity definition for binary and quaternary sensitive attributes, and generalized demographic parity for continuous sensitive attribute. Markers ■ ⬟ ✖ ◆ represent average prediction among specific discrete sensitive attributes, Red dashed line -- and blue solid line ∿ represent prediction average among all data and that with specific sensitive attribute. $TV(\cdot, \cdot)$ represents weighted total variation distance.

*unreliable* to adopt the mathematical bound of fairness metric to evaluate different algorithms since lower metrics bound does not necessarily imply lower prediction bias. A question is raised:

> ### *Can we extend DP for continuous attributes while preserving tractable computation?*

In this work, we provide positive answers via proposing *generalized demographic parity* (GDP) from regression perspective. Figure 1 provides an illustrative example for DP and GDP. The local prediction average (blue solid curve ∿) and the global prediction average (red dashed line --) represent the average prediction value given sensitive attributes and the whole data samples, respectively. The local and global prediction average should be consistent at any specific continuous sensitive attributes. Therefore, we define GDP, via the weighted total variation distance, to measure the distance between the local and global prediction average, where the weight is the probability density of continuous sensitive attributes. We also theoretically demonstrate the equivalence of GDP and DP for binary sensitive attributes, provide an understanding of GDP from probability perspective, and reveal the bias mitigation methods connection between GDP regularizer and adversarial debiasing.

Although GDP is clearly defined on the unknown underlying joint distribution of prediction and sensitive attributes, only data samples, in practice, are available. To this end, we propose two GDP estimation methods, named histogram estimation (hard group strategy) and kernel estimation (soft group strategy) methods, where kernel estimation is provable with faster estimation error convergence rate w.r.t. data sample size. Specifically, histogram estimation manually divides continuous sensitive attributes into several disjointed and complementary sensitive attributes bins, and then each sample only belongs to one specific bin containing the sensitive attribute of the sample. In other words, the group indicator of the sample is one-hot. As for kernel estimation, instead of quantizing continuous sensitive attributes as several groups, the group indicator of the sample is treated as a kernel function. In other words, to calculate the mean prediction value given specific sensitive attributes, group smoothing strategy is adopted via a soft indicator determined by the sensitive attribute distance between the sample sensitive attribute with target-specific sensitive attribute.

In short, the contributions of this paper are:

- We develop a tractable group fairness metric GDP for continuous sensitive attributes. We theoretically justify GDP via demonstrating the equivalence with DP for binary sensitive attributes, providing GDP understanding from probability perspective, and revealing the connection between GDP regularizer and adversarial debiasing.
- We propose histogram and kernel GDP estimation and provably demonstrate the superiority of kernel GDP estimation method with faster estimation error convergence rate w.r.t. sample size.
- We experimentally evaluate the effectiveness and expansibility of GDP on different domain benchmarks (e.g., tabular, graph, and temporal graph data), tasks (e.g, classification and regression tasks), and compositional sensitive attributes.

## 2 RELATED WORK

**Machine Learning Fairness** Fair machine learning targets bias mitigation for automated decision-making systems. Various fairness definitions, such as group fairness and individual fairness, have been proposed (Zemel et al., 2013). Group metrics, such as DP and EO, measure prediction dif-

ference between the groups with different sensitive attributes such as gender, age (Louizos et al., 2016; Hardt et al., 2016). While pre- and post-processing methods have been proposed for fairness boosting, these methods can still lead to higher prediction bias (Barocas et al., 2017) compared with in-processing methods, such as adding regularizer, adversarial debiasing and data augmentation. For example, the covariance between the predictions and sensitive attributes regularization are imposed to boost the independence in (Woodworth et al., 2017). (Zafar et al., 2017) constrains the decision boundaries of classifier to minimize prediction disparity between different groups. Adversarial training has been originally proposed for deep generative modeling (Goodfellow et al., 2014)and has been introduced for prediction debias in representation learning (Zhao et al., 2020; Beutel et al., 2017; Louppe et al., 2017) and transfer learning (Madras et al., 2018). Data augmentation, such as fair mixup (Chuang & Mroueh, 2020), can improve the generalization ability for fairness. Representation neutralization is proposed to boost fairness without sensitive attribute (Du et al., 2021).

**Kernel Density Estimation and Kernel Regression** Kernel density estimation (KDE) is a non-parametric method to estimate the continuous probability density function of a random variable (Davis et al., 2011; Parzen, 1962). Given finite data samples, KDE smoothly estimate the probability function via weighted summation, where the weight is determined via kernel function (Epanechnikov, 1969). Kernel regression is a non-parametric technique to estimate the conditional expectation of a random variable (Nadaraya, 1964). Nadaraya-Watson Kernel regression function estimator is proposed for regression via locally normalized weighted average in (Bierens, 1988), where the sample weight is determined by kernel function.

## 3    GENERALIZED DEMOGRAPHIC PARITY

Without loss of generality, we consider a binary classification task to predict the output variable $Y$ given the input variable $X$, while avoiding prediction bias for sensitive attribute $S$. Define the input $X \in \mathcal{X} \subset \mathbb{R}^d$, labels $Y \in \{0, 1\}$, and machine learning model $f : \mathbb{R}^d \to [0, 1]$ provides prediction score $\hat{Y} = f(X)$. Fairness requires predictor $\hat{Y}$ to be independent of sensitive attribute $S$, regardless of continuous or discrete, i.e., $P(\hat{Y} = \hat{y}) = P(\hat{Y} = \hat{y}|S = s)$ for any support value $y$ and $s$ (Beutel et al., 2017). Since the independent constraint is difficult to optimize, the relaxed demographic parity (DP) (Madras et al., 2018) metrics are proposed to quantitatively measure the predictor bias for binary sensitive attribute $S \in \{0, 1\}$. Formally, the demographic parity is defined as $\Delta DP = |\mathbb{E}_{\hat{Y}}[\hat{Y}|S = 0] - \mathbb{E}_{\hat{Y}}[\hat{Y}|S = 1]|$, where $\mathbb{E}[\cdot]$ represents variable expectation. For categorical sensitive attribute $S \in \mathcal{S}$, work (Cho et al., 2020) introduces a fairness metric, named difference w.r.t. demographic parity (DDP) $\Delta DDP = \sum_{s \in \mathcal{S}} |\mathbb{E}_{\hat{Y}}[\hat{Y}|S = s] - \mathbb{E}_{\hat{Y}}[\hat{Y}]|$.

Although DP has been widely used to evaluate the prediction bias, it is still inapplicable for continuous sensitive attributes since the data samples cannot be directly divided into several distinctive groups based on the sensitive attributes. Without loss of generality, we assume continuous sensitive attributes $S \in [0, 1]$ and propose GDP to extend tractable DP for continuous sensitive attributes. Assume the joint distribution of tuple $(S, \hat{Y})$ is $P_{S,\hat{Y}}(s, \hat{y})$, the local prediction average and global prediction average are defined as the prediction expectation given sensitive attribute $S = s$ and without any sensitive attribute condition, i.e., local prediction average $m(s) \triangleq \mathbb{E}[\hat{Y}|S = s]$ and global prediction average $m_{avg} \triangleq \mathbb{E}_S[m(S)] = \mathbb{E}[\hat{Y}]$, respectively. Then, we adopt weighted total variation distance on local prediction average and global prediction average, where the weight is specified by the probability density function of the sensitive attribute. The formal definition of the discrepancy demographic parity for continuous sensitive attributes is as follows:

$$\Delta GDP = \int_0^1 \left| m(s) - m_{avg} \right| P_S(S = s) \mathrm{d}s = \mathbb{E}_S[|m(S) - m_{avg}|], \tag{1}$$

We also provide the connection of GDP and DP for binary sensitive attributes, which implies that GDP is equivalent to DP for binary sensitive attributes, as follows:

**Theorem 1** (Connection between DP and GDP). *For binary sensitive attribute $S \in \{0, 1\}$, GPD and DP are equivalent except the coefficient only dependent on datasets. Specifically, the relation of $\Delta GDP$ and $\Delta DP$ satisfies $\Delta GDP = 2P_S(S = 1) \cdot P_S(S = 0) \cdot \Delta DP$.*

The proof of Theorem 1 is presented in Appendix A. For categorical sensitive attributes, it is easy to obtain that $\Delta GDP = \sum_{s \in \mathcal{S}} P_S(S = s)|\mathbb{E}_{\hat{Y}}[\hat{Y}|S = s] - \mathbb{E}_{\hat{Y}}[\hat{Y}]|$, i.e., GDP is weighted DDP for

categorical sensitive attributes. In a nutshell, GDP is a natural fairness metric extension for binary and categorical sensitive attributes. Since the independence between prediction $\hat{Y}$ and sensitive attribute $S$ implies that the joint distribution $P_{S,\hat{Y}}(s, \hat{y})$ and product marginal distribution $P_S(s)P_{\hat{Y}}(\hat{y})$ are the same, the bias can be measured by the distance of the joint distribution and product marginal distribution. Subsequently, we show the connection of GDP and prediction-weighted total variation distance between these two distributions as follows:

**Theorem 2** (Probability View of GDP). *Assume the joint distribution of* $(\hat{Y}, S)$ *with support* $[0, 1]^2$ *is* $P_{S,\hat{Y}}(s, \hat{y})$. *Define the prediction-weighted total variation distance as* $TV_{pred}(P^1, P^2) \triangleq \int_0^1 \int_0^1 \hat{y}|P^1(\hat{y}, s) - P^2(\hat{y}, s)|\mathrm{d}\hat{y}\mathrm{d}s$. *Then the proposed GDP for continuous senstive attribute is upper bounded by prediction- weighted total variation distance between the joint distribution and product marginal distribution:*

$$\Delta GDP = \int_0^1 \int_0^1 \left| \hat{y}\Big[ P_{S,\hat{Y}}(s, \hat{y}) - P_S(s)P_{\hat{Y}}(\hat{y}) \Big] \right| \mathrm{d}\hat{y}\mathrm{d}s \le TV_{pred}\Big( P_{S,\hat{Y}}(s, \hat{y}), P_S(s)P_{\hat{Y}}(\hat{y}) \Big).$$

The proof of Theorem 2 is presented in Appendix B. Theorem 2 demonstrates that GDP is actually a lower bound for prediction-weighted total variation distance between these two distributions and implies the necessity of GDP for bias measurement.

## 4 GDP ESTIMATION

GDP is defined based on the underlying joint distribution $P_{S,\hat{Y}}(s, \hat{y})$ of tuple $(S, \hat{Y})$, where $S, \hat{Y} \in [0, 1]$. The underlying joint distribution, however, is unknown. Thus, we aim to estimate GDP given samples $\{(s_n, \hat{y}_n), n \in [N]\}$, where $[N] \triangleq \{1, \cdots, N\}$. To bridge this gap, we propose histogram GDP estimation and kernel GDP estimation methods based on different group strategies. Specifically, histogram GDP estimation hardly groups data samples via creating consecutive, non-overlapping intervals bins, and the local prediction average is estimated by the average prediction among the data samples in the bin. As for kernel GDP estimation, a soft group indicator, determined by the kernel function and sensitive attribute distance, is adopted to provide group smoothing strategy. Specifically, the local prediction average for target sensitive attribute is calculated via weighted prediction, where the sample with sensitive attribute close to the target sensitive attribute possesses large weight.

**Histogram GDP Estimation** A histogram is originally an approximate representation of the underlying data probability distribution via creating several consecutive, non-overlapping intervals or bins with, usually but not required, equal bandwidth. In this paper, we assume all bins with equal bandwidth $h$ and the number of bins $N_h \triangleq \frac{1}{h}$ is an integer. In other words, the data tuples can be divided into $N_h$ groups, and the bin intervals are given by $B_1 = [0, h)$, $B_2 = [h, 2h)$, $\cdots$, $B_{N_h} = [(N_h - 1)h, 1]$. Define indicator function $\mathbb{I}(A)$ as 1 if event $A$ happens, otherwise is 0. Thereby, the group indicator $w_h(n, i)$ of sample $(s_n, \hat{y}_n)$ for $i - th$ bin is given by $\mathbb{I}(s_n \in B_i)$. Note that all bin intervals are complementary; each sample belongs one and only one bin, i.e., the indicator vectors $\mathbf{w}_h(n) = [w_h(n, 1), \cdots, w_h(n, N_h)]$ for sample $n$ is one-hot, which formally define the hard group strategy. The local prediction average and probability of sensitive attribute can be point-wisely estimated based on the empirical expectation and distribution. Specifically, for $s \in B_i$, the local and global prediction average are given by

$$\hat{m}^h\Big( s \in B_i \Big) = \sum_{n=1}^N \mathbb{I}(s_n \in B_i)\hat{y}_i, \text{ for } i \in [N_h]; \qquad \hat{m}_{avg}^h = \sum_{n=1}^N \hat{y}_i. \tag{2}$$

Similarly, the probability of sensitive attribute is given by $\hat{P}_S^h(s \in B_i) = \frac{\sum_{n=1}^N \mathbb{I}(s_n \in B_i)}{N}$. Finally, we combine all estimations to calculate prediction bias as follows:

$$\hat{\Delta} GDP(h) = \sum_{i=1}^{N_h} \left| \hat{m}^h\Big( s \in B_i \Big) - \hat{m}_{avg}^h \right| \hat{P}_S^h(s \in B_i). \tag{3}$$

**Kernel GDP Estimation** Histogram GDP estimation provides a hard group strategy via creating several bins. However, a tiny sensitive attribute perturbation can lead to different group indicators

and thus histogram estimation is not robust on sensitive attribute. On the other hand, a data tuple not necessarily belongs to one group for continuous attributes. For example, the data sample with $s = 0.5$ may have the same probability belonging to target sensitive attributes $s = 0.4$ and $s = 0.6$.

Based on these observations, we propose kernel GDP estimation via group smoothing. Intuitively, when calculating local prediction average or probability density function for target sensitive attribute, the tuple with more close attribute is entrusted higher weight. Specifically, we introduce a symmetric one-dimensional smoothing kernel function $K(s) \geq 0$ satisfying normalized condition $\int K(s)\mathrm{d}s = 1$, symmetry condition $\int sK(s)\mathrm{d}s = 0$ and finite variance $\sigma_K^2 \triangleq \int s^2 K(s)\mathrm{d}s > 0$. Define $h > 0$ as the kernel bandwidth. For target sensitive attribute $s$, the tuple weight for sample with sensitive attribute $s_n$ is given by $w(s_n, s) \triangleq \frac{1}{h}K(\frac{|s_n - s|}{h})$. In short, kernel function provides the group smoothing strategy based on sensitive attribute distance of tuple pair.

Given the smoothing kernel function $K(s)$, the local and global prediction average can be obtained via normalized weighted average (Nadaraya–Watson kernel estimator) as follows:

$$\tilde{m}^h(s) = \frac{\sum_{n=1}^{N} \hat{y}_n K(\frac{s_n - s}{h})}{\sum_{n=1}^{N} K(\frac{s_n - s}{h})}, \qquad \tilde{m}_{avg}^h = \frac{\sum_{n=1}^{N} \hat{y}_n}{N}. \tag{4}$$

Similarly, the probability of sensitive attribute is given by $\tilde{p}_S^h(s) = \frac{1}{Nh}\sum_{n=1}^{N} K(\frac{s_n - s}{h})$. Finally, we combine all estimations to calculate kernel GDP estimation as follows:

$$\tilde{\Delta}GDP(h) = \int_0^1 \left| \tilde{m}^h(s) - \tilde{m}_{avg}^h \right| \tilde{p}_S^h(s)\mathrm{d}s. \tag{5}$$

**Estimation Error Analysis** We provide the theoretical analysis on GDP estimation error and prove the superiority of kernel GDP estimation. Assume that each data sample is independent and identically distributed random variables. Therefore, the estimated GDP is still a random variable, and we adopt the expectation of the mean squared error (MSE) to quantify the accuracy of the estimation method. Formally, the error of histogram estimation and kernel estimation are given by

$$Err_{hist} = \mathbb{E}[|\hat{\Delta}GDP - \Delta GDP|^2]; \qquad Err_{kernel} = \mathbb{E}[|\tilde{\Delta}GDP - \Delta GDP|^2]. \tag{6}$$

where the expectation is taken across $N$ tuples. Here, we provide an asymptotic analysis on estimation error and show the superiority of kernel GDP estimation in the following:

**Theorem 3** (Estimation Error Convergence Rate). *Assume that the mean prediction function $m(s)$, given sensitive attribute $s$, is smooth and satisfies L- Lipschitz condition [2] on local average $|m(s) - m(s')| \leq L|s - s'|$ for any $s, s'$. the optimal bandwidth choice for histogram and kernel estimation methods are $h_{hist}^* = O(N^{-\frac{1}{3}})$ and $h_{kernel}^* = O(N^{-\frac{1}{5}})$. Additionally, the estimation error satisfy $Err_{hist} = O(N^{-\frac{2}{3}})$ and $Err_{kernel} = O(N^{-\frac{4}{5}})$, where $O(\cdot)$ is big O notation.*

The proof of Theorem 3 is presented in Appendix C. The proof sketch is to separately provide upper bounds for the estimation error of local average and sensitive attribute probability density estimation. Finally, we combine these two estimations to obtain the estimation error for GDP.

**Computation Complexity Analysis** Since GDP calculation involves in integral operations, the approximated numerical integration is usually adopted with $M$ probing sensitive attribute. The complexity to calculate local prediction average and probability density at $M$ probing sensitive attributes are $O(MN)$ and thus, the complexity for histogram and kernel estimation both are $O(MN)$. In (Mary et al., 2019), intractable HGR coefficient equals second large singular value of distribution ratio matrix or can be upper bounded by tractable chi-square distance between the joint distribution and marginal product distribution. Assume that there are $M$ probing prediction, the complexity for two-dimensional probability density function estimation and SVD is $O(M^2N)$ and $O(M^3)$, and that of the chi-square distance is $O(M^2N)$. Therefore, the computation complexity for HGR is $O(M^2(M + N))$. As for the neural based approximation for HGR or mutual information, the complexity for training is quite large compared with directly computation.

---

[2]Lipschitz condition requires bounded gradient w.r.t. sensitive attribute and guarantees the rationale local average estimation based on neighbor data samples.

## 5 ANALYSIS ON GDP REGULARIZER AND ADVERSARIAL DEBIASING

With GDP bias measurement for continuous sensitive attributes, it is natural to add GDP regularizer to enforce fairness. Another bias mitigation is adversarial debiasing, a two-player framework for predictor and adversary with regression task. We establish the connection between GDP regularizer and adversarial debiasing, and demonstrate that *adversarial debiasing with specific adversary regression objective is actually minimizing GDP implicitly.*

**GDP Regularizer:** As a reminder, our goal is to learn a predictor $\hat{Y} = f(X)$ that approximates the label $Y$ for each input while reducing the prediction bias $\Delta GDP$ w.r.t. continuous sensitive attributes $S$. It is natural to add GDP regularization to enforce fairness. Given the prediction loss $\mathcal{L}_{pred}$, the fairness-enforcing objective function is $\min_f \mathcal{L}_{pred}\Big(f(X), Y\Big) + \lambda \Delta GDP$, where $\mathcal{L}$ could be regression or classification task loss and $\lambda$ is the hyperparameter to control the trade-off between the prediction performance and prediction bias reduction.

**Adversarial Debiasing:** Adversarial debiasing is another natural method to ensure fair prediction via a two-player game between predictor and adversary. Specifically, the predictor $f$ yields prediction $\hat{Y}$, and given prediction $\hat{Y}$, the adversary $g$ tries to predict continuous sensitive attributes $\hat{S} = g(\hat{Y})$ in regression task. Similar to adversarial debiasing (Louppe et al., 2017), we adopt the same classifier and adversary structure, where the input of adversary is the output of the classifier. For objective function of adversary, we adopted the utility function proposed in (Madras et al., 2018) [3] to represent sensitive attribute prediction accuracy. In this case, the predictor aims to generate accurate prediction and fool the adversarial simultaneously, while adversary targets high utility for accurate sensitive attribute prediction. Let $\mathcal{L}_{pred}$ denote the prediction loss and $\mathcal{L}_{adv}$ represent the adversarial utility. Then adversarial debiasing is trained following the min-max procedure $\min_f \max_g \mathcal{L}_{pred}\Big(f(X), Y\Big) + \lambda \mathcal{L}_{adv}\Big(g(f(X)), S\Big)$. Define $g^* = \arg\min_g \mathcal{L}_{adv}\Big(g(f(X)), S\Big)$ as the optimal adversarial given predictor $f$, the min-max procedure is simplified to the objective function $\min_f \mathcal{L}_{pred}\Big(f(X), Y\Big) + \lambda \mathcal{L}_{adv}\Big(g^*(f(X)), S\Big)$.

**Theoretical Connection:** We provide an inherent connection between GDP regularizer and adversarial debiasing. Specifically, we demonstrate that the optimal adversarial utility $\mathcal{L}_{adv}\Big(g^*(f(X)), S\Big)$ is actually the upper bound of GDP $\Delta GDP$ as long as the utility function in adversary is $\mathcal{L}_{adv}(\hat{S}, S) = 1 - |\hat{S} - S|$ in the following theorem:

**Theorem 4** (GDP and Adversarial Debiasing Connection). *Considering a predictor $\hat{Y} = f(X)$ and adversary $\hat{S} = g(\hat{Y})$, given adversary utility $\mathcal{L}_{adv}(\hat{S}, S) = 1 - |\hat{S} - S|$ and optimal adversary $g^*$, then GDP $\Delta GDP$ is bounded by the utility function $\mathcal{L}_{adv}\Big(g^*(f(X)), S\Big)$ with optimal adversary, i.e., $\mathcal{L}_{adv}\Big(g^*(f(X)), S\Big) \geq \Delta GDP$.*

The proof of Theorem 4 is presented in Appendix E. Intuitively, a fair classifier aims to minimize the adversary utility to make prediction fair, and thus induces lower GDP metric. Theorem 4 reveals the inherent connection between GDP regularizer and adversarial debiasing: adversarial debiasing behaviors like predictor loss optimization with GDP regularization, as long as the adversary is optimal. Additionally, such connection not only holds for underlying data distribution, but also for empirical data distribution. The proof sketch is similar via replacing underlying data distribution as empirical data distribution in the expectation operation. In practice, the alternative optimization is usually adopted in adversarial debiasing via alternatively updating either predictor or adversary at each training step while keeping the other one fixed.

---

[3] Work (Madras et al., 2018) adopts encoder-decoder structure for transferable representation learning. We only adopt utility function to connect GDP and adversarial debiasing (Louppe et al., 2017).

## 6 EXPERIMENTS

We evaluate the effectiveness and expansibility of *GDP*. First, we show the lower GDP estimation error for kernel GDP estimation [4] with group smoothing, compared to that of histogram, via two synthetic experiments. We empirically evaluate the effectiveness and expansibility of kernel estimation for multiple prediction tasks, including classification and regression tasks, and multiple domain real-world datasets, including tabular, graph and temporal graph data (See Appendix H.1). Moreover, the kernel estimation is also applicable for compositional continuous sensitive attributes. For a fair comparison, we compare our method **kernel**, adding kernel GDP estimation as regularization, with (a) *vanilla*: training with empirical risk minimization (ERM) without any regularization; (b) *histogram*: histogram estimation with continuous sensitive attribute; (c) *adv*: adversarial debiasing (Louppe et al., 2017); (d) *adv-bn*: adversarial debiasing with binary-quantized sensitive attribute; (e) *hgr*: the upper bound of HGR as regularizer (Mary et al., 2019); and (f)*hgr-bn*: the upper bound of HGR as regularizer with binary-quantized sensitive attribute. Specifically, we demonstrate the trade-off between prediction performance and GDP by varying the hyper-parameter $\lambda$. In particular, we adopt accuracy (Acc) for classification task and mean absolute error (MAE) for regression task. Details about the baselines and experimental setups for each dataset can be found in Appendix H.2.

### 6.1 SYNTHETIC EXPERIMENTS

We test GDP estimation error and investigate the robustness for histogram and kernel estimation via two synthetic experiments. The goal is to investigate the effect of bandwidth choice and number of samples on GDP estimation error. **For data generation**, we first generate bivariate Gaussian distribution $\mathcal{N}(\mu, \boldsymbol{\Sigma})$ for samples $(S, \hat{Y})$, where expectation $\mu = [0.5, 1.0]$ and covariance matrix $\boldsymbol{\Sigma} = \begin{bmatrix} 1 & 0.5 \\ 0.5 & 2.0 \end{bmatrix}$. Additionally, we generate the samples based on $p_{S,\hat{Y}}(s, \hat{y}) = s + \hat{y}$ if $0 \leq s, \hat{y} \leq 1$, via acceptance-rejection method (Wells et al., 2004). **For estimation**, we select two typical kernel functions (tricube and Aitchison-Aitken kernel (Mussa, 2013)) for local prediction average, and two kernel functions (linear and cosine kernel functions) for probability density function estimation.

Figure 2 shows the GDP estimation error w.r.t. bandwidth $h$ and number of samples $N$ for bivariate Gaussian distribution and second synthetic bivariate distribution, respectively. We observe that histogram GDP estimation error is highly sensitive to bandwidth choice, while kernel estimation is robust to bandwidth and kernel function choice due to the flexibility of group smoothing. As for error rate with number of samples, our experiments show the error rate curve for histogram and kernel estimation. It is seen that kernel estimation can achieve the fast error convergence rate if searching the optimal bandwidth for each method. In addition, the estimation error result is almost the same for different kernel functions, which supports the theoretical results in Theorem 3.

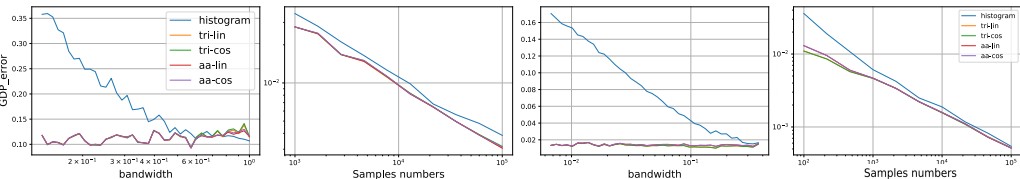

Figure 2: Demographic parity estimation error analysis with respect to bandwidth and number of samples for Gaussian distribution and second synthetic distributions.

### 6.2 EXPERIMENTS ON TABULAR DATA

**Dataset:** We consider two benchmark tabular datasets, UCI Adult and Crimes, to evaluate the effectiveness of kernel estimation for classification and regression tasks. UCI Adult dataset [5] contains more than $40,000$ individual information from 1994 US Census. The classification task is to predict

---

[4]Since there is not ground truth on underlying data distribution, we adopt reliable kernel-based estimated GDP as fairness metric in real-world datasets.

[5]https://archive.ics.uci.edu/ml/datasets/adult

whether a person's income exceeds $50k/yr$ (KOHAVI, 1996). We consider normalized age [6] sensitive attribute to measure the fairness of algorithms. The Crime dataset [7] includes 128 attributes for $1,994$ samples from communities in the US. The regression task is to predict the number of violent crimes per population in US communities. We adopt the black group ratio as continuous sensitive attribute. **Model:** We adopt two-layer selu networks model (Klambauer et al., 2017) with hidden size 50 and report the mean prediction performance and GDP with 5 running times.

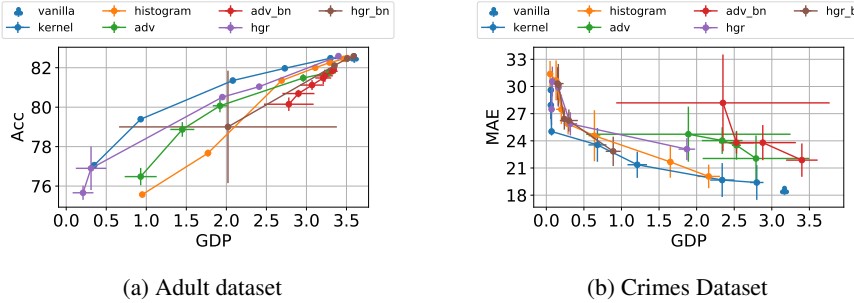

(a) Adult dataset                     (b) Crimes Dataset

Figure 3: Mitigation performance on tabular dataset with kernel estimation and other baselines. (a) The tradeoff between Accuracy (classification) and GDP for Adult dataset; (b) The tradeoff between MAE (regression) and GDP for Crimes dataset.

**Results:** We compare the bias mitigation performance, i.e., the tradeoff between prediction performance and GDP, of kernel estimation with other baselines for the two tabular datasets in Figure 3. The hyper-parameter $\lambda$ in Eq. (5) controls the tradeoff between prediction performance and GDP. Specifically, we choose accuracy metric for classification task in Adult dataset, and MAE metric for regression task in Crimes datset. For adversarial debiasing, we vary the regularization weights to obtain the tradeoff curve. **Overall, we make the following observations**: **(a)** kernel estimation outperforms all other baseline methods in terms of performance-fairness tradeoff curve for Adult and Crimes datasets. Specifically, kernel estimation can achieve more than $1\%$ accuracy improvement if GDP is lower than $0.02$ for adult dataset, and more than $2\%$ MAE reduction for Crimes dataset; **(b)** kernel estimation has lower computation complexity compared with adversarial debiasing. Kernel estimation and histogram decrease more than $50\%$ running time on an average across all tabular dataset, which makes kernel and histogram estimation readily usable for large scale real-world datasets; **(c)** The histogram estimation and binary-quatized sensitive attribute would lead to mitigation performance drop for kernel estimation, adversarial debiasing and HGR. This fact implies the importance of order information in continuous sensitive attributes for bias mitigation. In addition, we observe adversarial debiasing training and HGR are not stable and large hyper-parameter unnecessarily leads to bias mitigation.

## 6.3 EXPERIMENTS ON GRAPH DATA

**Dataset**: We consider two real-world graph datasets, *Pokec-z* and *Pokec-n*, sampled from a larger one Facebook-like social network Pokec in Slovakia. User profiles contain gender, age, interest, education, working field and etc. We treat the normalized age as the continuous sensitive attributes and the node classification task is to predict the working field of the users. **Model**: We use three graph neural network backbones, graph convolutional networks (GCN) (Kipf & Welling, 2017), graph attention networks (GAT) (Veličković et al., 2018) and Simplifying graph convolutional networks (SGC) (Wu et al., 2019) with 64 feature dimensions. We train GNN with 200 epochs with 5 running times and report the average accuracy and GDP. In each trial, the dataset is randomly split into a training, validation, and test set with $50\%$, $25\%$, and $25\%$ partition, respectively.

**Results:** We compare the mitigation performance of kernel estimation with other baselines for two datasets with three backbones in Figure 4. Similarly, kernel estimation consistently outperforms the other baselines by a large margin and binary-quantized sensitive attributes inevitably deteriorate the mitigation performance. Another observation is that GDP can be reduced at least $80\%$ at the cost

---

[6]The biggest difference between continuous and discrete sensitive attribute falls in the existence of order information (Mary et al., 2019) for attribute values.

[7]https://archive.ics.uci.edu/ml/datasets/communities+and+crime

of 2% accuracy for kernel estimation in two datasets and three backbones, while results in larger accuracy drop for other baselines. Additionally, adversarial debiasing and HGR are also unstable during training and higher hyper-parameter may lead to larger GDP.

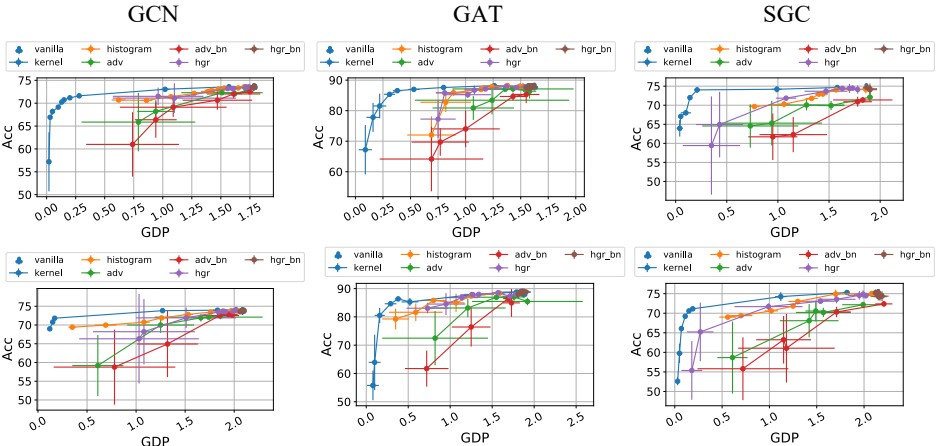

Figure 4: Mitigation performance of GCN, GAT and SGC for Pokec-n and Pokec-z dataset.

### 6.4 Experiments on Compositional Continuous Sensitive Attributes

**Dataset**: Similarly, the same two benchmark tabular datasets are adopted to evaluate the effectiveness of kernel estimation for compositional continuous sensitive attributes. Specifically, we treat the normalized age and education number for UCI dataset, and black group ratio and normalized age for Crimes dataset as compositional continuous sensitive attributes. to evaluate bias mitigation performance. **Model**: We also adopt two-layer selu networks model with hidden size 50 with 5 running times and report the average mean prediction performance and GDP.

**Results**: Figure 5 shows bias mitigation performance between prediction performance and GDP. Again, kernel estimation consistently achieves a better tradeoff compared with all other baselines, and binary-quantized compositional sensitive attribute leads to mitigation performance drop.

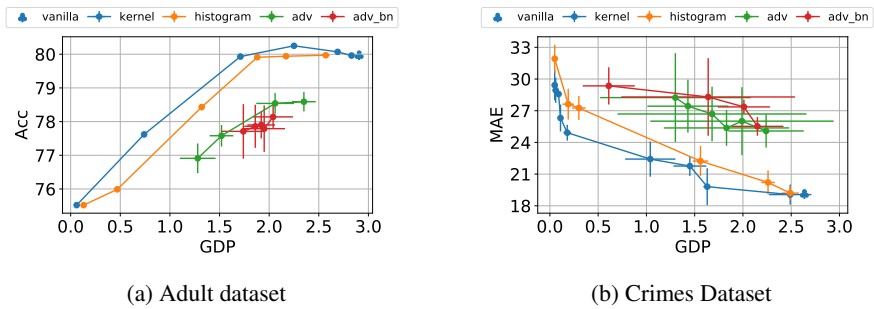

(a) Adult dataset        (b) Crimes Dataset

Figure 5: Mitigation performance for compositional sensitive attributes.

## 7 Conclusion

We generalize demographic parity fairness metric, named GDP, to continuous sensitive attributes while preserving tractable computation. We theoretically justify the unification of proposed GDP for continuous and discrete sensitive attributes, and show the necessity of GDP via demonstrating the connection with joint and product margin distributions distance. We also propose two GDP estimation methods, named histogram and kernel, with linear computation complexity via hard and soft group strategies, and provide corresponding estimation error analysis. For the superiority of kernel estimation, we provably demonstrate the faster estimation error convergence rate compared with histogram estimation, and experimentally show better bias mitigation performances in multiple domains, multiple tasks and compositional sensitive attributes.

## 8 ACKNOWLEDGEMENTS

We would like to sincerely thank everyone who has provided their generous feedback for this work. Thank the anonymous reviewers for their thorough comments and suggestions. This work was supported in part by X-Grant project, National Science Foundation IIS-1939716, IIS-1900990, IIS-1750074 grant, and JPMorgan Faculty Research Award.

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

## A  PROOF OF THEOREM 1

For binary sensitive attribute $S \in \{0,1\}$, the probability of sensitive attribute follows Bernoulli distribution $\Big( P_S(S=0), P_S(S=1) \Big)$. Therefore, the global prediction average is given by $m_{avg} = P_S(S=0)m(0) + P_S(S=1)m(1)$ and GDP is

$$
\begin{aligned}
\Delta GDP &= P_S(S=0)\Big|m(0) - m_{avg}\Big| + P_S(S=1)\Big|m(1) - m_{avg}\Big| \\
&= P_S(S=0)\Big|m(0) - P_S(S=0)m(0) - P_S(S=1)m(1)\Big| \\
&\quad + P_S(S=1)\Big|P_S(S=0)m(0) + P_S(S=1)m(1) - m(1)\Big| \\
&= 2P_S(S=0)P_S(S=1)|m(0) - m(1)| \\
&= 2P_S(S=0)P_S(S=1)\Delta DP.
\end{aligned}
$$

where the coefficient $2P_S(S=0)P_S(S=1)$ only depends on data. This fact demonstrates the applicability of GDP for categorical sensitive attribute with discrete measure choice for sensitive attributes, and it is equivalent to demographic parity for binary sensitive attribute.

## B  PROOF OF THEOREM 2

Notice that the fairness constraint ideally requires the independence of prediction $\hat{Y}$ and sensitive attribute $S$, i.e., the joint distribution and product margin distribution equals: $P_{\hat{Y},S}(\hat{y}, s) = P_S(s)P_{\hat{Y}}(\hat{y})$. Aiming to measure independence deviation, a natural intuition is to quantify the probability deviation via the distance between the joint distribution and product margin distribution. We show the connection between the proposed GDP and prediction-weighted total variation distance of these two distributions. Recall the definition of GDP, it is easy to obtain

$$
\begin{aligned}
\Delta GDP &= \int_0^1 \Big|m(s) - m_{avg}\Big| P_S(S=s)\mathrm{d}s \\
&= \int_0^1 \Big| \int_0^1 \hat{y}P_{\hat{Y}|S}(y|s)\mathrm{d}\hat{y} - \int_0^1 \hat{y}P_{\hat{Y}}(y)\mathrm{d}\hat{y} \Big| P_S(S=s)\mathrm{d}s \\
&= \int_0^1 \Big| \int_0^1 \hat{y}\Big( P_{\hat{Y},S}(y,s) - P_S(s)P_{\hat{Y}}(\hat{y}) \Big)\mathrm{d}\hat{y}\Big| \mathrm{d}\hat{y}\mathrm{d}s \\
&\leq \int_0^1 \int_0^1 \hat{y}\Big|P_{\hat{Y},S}(y,s) - P_S(s)P_{\hat{Y}}(\hat{y})\Big| \mathrm{d}\hat{y}\mathrm{d}s \\
&= TV_{\hat{y}}(P_{\hat{Y},S}(\hat{y},s), P_S(s)P_{\hat{Y}}(\hat{y}).
\end{aligned}
$$

which completes the proof.

## C  PROOF OF THEOREM 3

Notice that histogram and kernel estimation require local prediction average and probability density function estimation for sensitive attributes, we start with the analysis on local prediction average and probability density function estimation, and then provide the proof of GDP error analysis.

### C.1  PROOF FOR HISTOGRAM ESTIMATION

Recall that the number of bins is $N_h$ with same bandwidth $h$, and bins interval are given by $B_1 = [0, h), B_2 = [h, 2h), \cdots, B_{N_h} = [(N_h - 1)h, 1]$. Firstly, we will separately analyze the error for local prediction average and sensitive attribute probability. Next, these two estimation error results can be combined for GDP estimation error.

Since the continuous sensitive attributes is considered, we defined the probability density function of sensitive attribute $S$ as $p_S(s)$. The estimated probability density function is given by:

$$\hat{p}_S^h(s) = \frac{1}{h}\hat{P}_S^h(s \in B_i) = \frac{1}{Nh}\sum_{n=1}^{N}\mathbb{I}(s_n \in B_i) \text{ for } s \in B_i; \tag{7}$$

Subsequently, we define the pointwise MSE error $MSE_{hist}^{pdf}(s)$ to measure probability density function estimation error given sensitive attribute $s$ as follows:

$$MSE_{hist}^{pdf}(s) = \mathbb{E}[|\hat{p}_S^h(s) - p_S(s)|^2], \tag{8}$$

where the expectation is took across $N$ samples. Then we have following Lemma 1 on optimal pointwise MSE error for probability density function:

**Lemma 1.** *Assume the mean prediction function $m(s)$, given sensitive attribute $s$, is smooth and satisfies L-Lipschitz condition $|m(s) - m(s^{'})| \leq L|s - s^{'}|$ for any $s, s^{'}$. Given $N$ i.i.d. samples $\{(\hat{y}_n, s_n), n \in [N]\}$, then the optimal MSE error $\min_h MSE_{hist}^{pdf}(s)$ is $O(N^{-\frac{2}{3}})$, where the optimal bandwidth satisfies $h^* = O(N^{-\frac{1}{3}})$ for any sensitive attribute $s$.*

As for the local prediction average, similarly, we have the estimated local prediction average as follows:

$$\hat{m}(s) = \frac{\sum_{n=1}^{N}\mathbb{I}(s_n \in B_i)\hat{y}_n}{\sum_{n=1}^{N}\mathbb{I}(s_n \in B_i)} \text{ for } s \in B_i; \tag{9}$$

Subsequently, we also define the pointwise MSE error $MSE_{hist}^{reg}(s)$ to measure local prediction average estimation error given sensitive attribute $s$ as follows:

$$MSE_{hist}^{reg}(s) = \mathbb{E}[|\hat{m}^h(s) - m(s)|^2], \tag{10}$$

where the expectation is took across $N$ samples. Then we have following Lemma 2 on optimal pointwise MSE error for local prediction average:

**Lemma 2.** *Assume the mean prediction function $m(s)$, given sensitive attribute $s$, is smooth and satisfies L-Lipschitz condition $|m(s) - m(s^{'})| \leq L|s - s^{'}|$ for any $s, s^{'}$. Define the bounded prediction variance $\sigma^2(s) \leq \sigma^2$, given sensitive attribute $s$, as $\sigma^2(s) \triangleq \mathbb{E}_{\hat{Y}|S}[(\hat{Y} - m(s))^2 | S = s]$. Given $N$ i.i.d. samples $\{(\hat{y}_n, s_n), n \in [N]\}$, then the optimal MSE error $\min_h MSE_{hist}^{reg}(s)$ is $O(N^{-\frac{2}{3}})$, where the optimal bandwidth satisfies $h^* = O(N^{-\frac{1}{3}})$ for any sensitive attribute $s$.*

**Proof for histogram estimation**: Based on the definition of MSE error in Eq. (6), we have

$$
\begin{aligned}
Err_{hist} &= \mathbb{E}[|\hat{\Delta}GDP - \Delta GDP|^2] \\
&\overset{(a)}{\leq} \mathbb{E}\Big[\Big\{\int_0^1 \Big||\hat{m}^h(s) - \hat{m}_{avg}|\hat{p}_S^h(s) - |m(s) - m_{avg}|p_S(s)\Big|\mathrm{d}s\Big\}^2\Big] \\
&\overset{(b)}{\leq} \mathbb{E}\Big[\Big\{\int_0^1 \Big||\hat{m}^h(s) - \hat{m}_{avg}| \cdot |\hat{p}_S^h(s) - p_S(s)| - |m(s) - m_{avg} - \hat{m}^h(s) + \hat{m}_{avg}| \cdot p_S(s)\Big|\mathrm{d}s\Big\}^2\Big] \\
&\overset{(c)}{\leq} 2\mathbb{E}\Big[\int_0^1 |\hat{m}^h(s) - \hat{m}_{avg}|^2\mathrm{d}s\Big] \cdot \mathbb{E}\Big[\int_0^1 |\hat{p}_S^h(s) - p_S(s)|^2\mathrm{d}s\Big] \\
&\quad + 2*2\Big\{\mathbb{E}\Big[\int_0^1 |\hat{m}^h(s) - m(s)|^2 p_S(s)\mathrm{d}s + \mathbb{E}\Big[\int_0^1 |\hat{m}_{avg}^h - m_{avg}|^2 p_S(s)\mathrm{d}s\Big]\Big\} \\
&\overset{(d)}{\leq} 2*O(N^{-\frac{2}{3}}) + 4*O(N^{-\frac{2}{3}}) + 4*N^{-1} = O(N^{-\frac{2}{3}}).
\end{aligned}
$$

where inequality $(a)$ holds due to absolute value inequality, inequality $(b)$ holds due to $|a_1 b_1 - a_2 b_2| \leq |a_1||b_1 - b_2| + |a_1 - a_2||b_2|$, inequality $(c)$ holds based on $(a + b)^2 \leq 2(a^2 + b^2)$ and Cauchy–Schwarz inequality, inequality $(d)$ holds based on Lemmas (1) and (2). Note that the order of the optimal bandwidth are the same in Lemmas (1) and (2), the optimal bandwidth for minimizing GDP MSE is $O(N^{-\frac{1}{3}})$.

## C.2 KERNEL ESTIMATION

Recall that we assume that kernel function satisfies normalized condition $\int K(s)\mathrm{d}s = 1$, symmetry $\int sK(s)\mathrm{d}s = 1$ and finite variance $\sigma_K^2 \triangleq \int s^2 K(s)\mathrm{d}s > 0$. We also define $\tilde{\sigma}_K^2 = \int K^2(y)\mathrm{d}y$. Similarly, the probability density function of sensitive attribute is given by $p_S(s)$. The estimated probability density function is given by:

$$\tilde{p}_S^h = \frac{1}{Nh} \sum_{n=1}^{N} K(\frac{s_n - s}{h}), \tag{11}$$

Subsequently, we define the pointwise MSE error $MSE_{kernel}^{pdf}(s)$ to measure probability density function estimation error given sensitive attribute $s$ as follows:

$$MSE_{hist}^{pdf}(s) = \mathbb{E}[|\tilde{p}_S^h(s) - p_S(s)|^2], \tag{12}$$

where the expectation is took across $N$ samples. Then we have following Lemma 3 on optimal pointwise MSE error for probability density function:

**Lemma 3.** *Given $N$ i.i.d. samples $\{(\hat{y}_n, s_n), n \in [N]\}$, then the optimal MSE error $\min_h MSE_{kernel}^{pdf}(s)$ is $O(N^{-\frac{4}{5}})$, where the optimal bandwidth satisfies $h^* = O(N^{-\frac{1}{5}})$ for any sensitive attribute $s$.*

As for the local prediction average, similarly, we have local prediction average estimation as follows:

$$\tilde{m}(s) = \frac{\sum_{n=1}^{N} K(\frac{s_n - s}{h})\hat{y}_n}{\sum_{n=1}^{N} K(\frac{s_n - s}{h})}, \tag{13}$$

Subsequently, we also define pointwise MSE error $MSE_{kernel}^{reg}(s)$ to measure local prediction average estimation error given sensitive attribute $s$ as follows:

$$MSE_{kernel}^{reg}(s) = \mathbb{E}[|\hat{m}^h(s) - m(s)|^2], \tag{14}$$

where the expectation is took across $N$ samples. Then we have following Lemma 4 on optimal pointwise MSE error for local prediction average:

**Lemma 4.** *Define the bounded prediction variance $\sigma^2(s) \leq \sigma^2$, given sensitive attribute $s$, as $\sigma^2(s) \triangleq \mathbb{E}_{\hat{Y}|S}[(\hat{Y} - m(s))^2|S = s]$. Given $N$ i.i.d. samples $\{(\hat{y}_n, s_n), n \in [N]\}$, then the optimal MSE error $\min_h MSE_{kernel}^{reg}(s)$ is $O(N^{-\frac{4}{5}})$, where the optimal bandwidth satisfies $h^* = O(N^{-\frac{1}{5}})$ for any sensitive attribute $s$.*

**Proof for kernel estimation**: Based on the definition of MSE error in Eq. (6), we have

$$
\begin{aligned}
Err_{kernel} &= \mathbb{E}[|\tilde{\Delta}GDP - \Delta GDP|^2] \\
&\overset{(e)}{\leq} \mathbb{E}\Big[\Big\{\int_0^1 \Big||\tilde{m}^h(s) - \tilde{m}_{avg}|\tilde{p}_S^h(s) - |m(s) - m_{avg}|p_S(s)\Big|\mathrm{d}s\Big\}^2\Big] \\
&\overset{(f)}{\leq} \mathbb{E}\Big[\Big\{\int_0^1 \Big||\tilde{m}^h(s) - \tilde{m}_{avg}| \cdot |\tilde{p}_S^h(s) - p_S(s)| - |m(s) - m_{avg} - \tilde{m}^h(s) + \tilde{m}_{avg}| \cdot p_S(s)\Big|\mathrm{d}s\Big\}^2\Big] \\
&\overset{(g)}{\leq} 2\mathbb{E}\Big[\int_0^1 |\tilde{m}^h(s) - \tilde{m}_{avg}|^2\mathrm{d}s\Big] \cdot \mathbb{E}\Big[\int_0^1 |\tilde{p}_S^h(s) - p_S(s)|^2\mathrm{d}s\Big] \\
&\quad + 2*2\Big\{\mathbb{E}\Big[\int_0^1 |\tilde{m}^h(s) - m(s)|^2 p_S(s)\mathrm{d}s + \mathbb{E}\Big[\int_0^1 |\tilde{m}_{avg}^h - m_{avg}|^2 p_S(s)\mathrm{d}s\Big]\Big\} \\
&\overset{(h)}{\leq} 2*O(N^{-\frac{4}{5}}) + 4*O(N^{-\frac{4}{5}}) + 4*N^{-1} = O(N^{-\frac{4}{5}}).
\end{aligned}
$$

where inequality $(e)$ holds due to absolute value inequality, inequality $(f)$ holds due to $|a_1 b_1 - a_2 b_2| \leq |a_1||b_1 - b_2| + |a_1 - a_2||b_2|$, inequality $(g)$ holds based on $(a + b)^2 \leq 2(a^2 + b^2)$ and Cauchy–Schwarz inequality, inequality $(h)$ holds based on Lemmas (3) and (4). Note that the order of the optimal bandwidth are the same in Lemmas (3) and (4), the optimal bandwidth for minimizing GDP MSE is $O(N^{-\frac{1}{5}})$.

# D PROOF OF LEMMAS

## D.1 PROOF OF LEMMA 1

*Proof.* Note that there exists bias-variance tradeoff for $MSE_{hist}^{pdf}(s)$, i.e.,

$$
\begin{aligned}
MSE_{hist}^{pdf}(s) &= \mathbb{E}[|\hat{p}_S^h(s) - p_S(s)|^2] \\
&= \underbrace{\left|\mathbb{E}[\hat{p}_S^h(s)] - p_S(s)\right|^2}_{Bias_{hist}^{pdf}(s)} + \underbrace{\mathbb{E}[\left|\hat{p}_S^h(s) - \mathbb{E}[\hat{p}_S^h(s)]\right|^2]}_{Var_{hist}^{pdf}(s)},
\end{aligned}
$$

Next we analyze the bias and variance for probability density function MSE. For the bias part, note that, for $s \in B_i$, the expectation of estimated probability density function satisfies:

$$
\mathbb{E}[\hat{p}_S^h(s)] = \frac{1}{h} P(s_n \in B_i) = \frac{\int_{(i-1)h}^{ih} p_S(s) \mathrm{d}s}{h} = p_S(s^*), \quad s^* \in B_i;
$$

where the last equality holds by the mean value theorem. Therefore, the bias satisfies

$$
\begin{aligned}
Bias_{hist}^{pdf}(s) &= \left|\mathbb{E}[\hat{p}_S^h(s)] - p_S(s)\right| \\
&\leq L|s^* - s| \leq Lh.
\end{aligned}
$$

As for variance part, note that the variance of Bernoulli distribution with parameter $p$ is $p(1-p)$, for $s \in B_i$, we have the variance as follows,

$$
\begin{aligned}
Var_{hist}^{pdf}(s) &= \frac{1}{h^2}\mathbb{D}[\frac{1}{N}\sum_{n=1}^{N}\mathbb{I}(s \in B_i)] = \frac{P(s_n \in B_i)[1 - P(s_n \in B_i)]}{Nh^2} \\
&\leq \frac{hp_S(s^*)}{Nh^2} = \frac{p_S(s^*)}{Nh};
\end{aligned}
$$

Combining the bias and variance part, we have

$$
MSE_{hist}^{pdf}(s) = [Bias_{hist}^{pdf}(s)]^2 + Var_{hist}^{pdf}(s) \leq L^2 h^2 + \frac{p_S(s^*)}{Nh}, \tag{15}
$$

It is easy to obtain the optimal bandwidth $h^* = [\frac{p_S(s^*)}{2L^2 N}]^{\frac{1}{3}} = O(N^{-\frac{1}{3}})$, and the minimized MSE is lower than $[\frac{Lp_s(s^*)}{2N}]^{\frac{2}{3}} = O(N^{-\frac{2}{3}})$. $\qquad\square$

## D.2 PROOF OF LEMMA 2

*Proof.* Note that the local prediction average by histogram, for $s \in B_i$, is given by $\hat{m}(s) = \frac{\sum_{n=1}^{N}\mathbb{I}(s_n \in B_i)Y_n}{\sum_{n=1}^{N}\mathbb{I}(s_n \in B_i)}$. Define the normalized weight as $w_n(s_n) = \frac{\mathbb{I}(s_n \in B_i)}{\sum_{n=1}^{N}\mathbb{I}(s_n \in B_i)}$, then local prediction average is given by $\hat{m}(s) = \sum_{n=1}^{N} w_n(s_n)m(s_n)$. Similarly, we can obtain the bias and variance tradeoff for local prediction average error $MSE_{hist}^{reg}(s)$ as follows,

$$
\begin{aligned}
MSE_{hist}^{reg}(s) &= \mathbb{E}[|\hat{m}^h(s) - m(s)|^2] \\
&= \underbrace{\left|\mathbb{E}[\hat{m}^h(s)] - m(s)\right|^2}_{Bias_{reg}^{hist}(s)} + \underbrace{\mathbb{E}[\left|\hat{m}^h(s) - \mathbb{E}[\hat{m}^h(s)]\right|^2]}_{Var_{reg}^{hist}(s)},
\end{aligned}
$$

For the bias part, based on Lipschitz condition on the mean prediction function, note that $\sum_{n=1}^{N} w_n(s_n) = 1$, we have

$$
\begin{aligned}
Bias_{reg}^{hist}(s) &= \left|\mathbb{E}[\hat{m}^h(s)] - m(s)\right| = \left|\sum_{n=1}^{N} w_n(s_n)[m(s_n) - m(s)]\right| \\
&\leq \sum_{n=1}^{N} w_n(s_n)\left|[m(s_n) - m(s)]\right| \leq \sum_{n=1}^{N} w_n(s_n)Lh = Lh. \tag{16}
\end{aligned}
$$

For the variance part, we have variance for local prediction average as follows,

$$
\begin{aligned}
Var_{reg}^{hist}(s) &= \mathbb{D}[\sum_{n=1}^{N} w_n(s_n)\hat{y}_n] = \mathbb{E}\Big[[\frac{\sum_{n=1}^{N} \mathbb{I}(s_n \in B_i)(Y_n - m(s_n))}{\sum_{n=1}^{N} \mathbb{I}(s_n \in B_i)}]^2\Big] \\
&\leq \sum_{n=1}^{N} \mathbb{E}\Big[\frac{\mathbb{I}(s_n \in B_i)\sigma^2}{\Big(\sum_{n=1}^{N} \mathbb{I}(s_n \in B_i)\Big)^2}\Big] \leq \frac{N\sigma^2/N_h}{(N/N_h)^2} = \frac{\sigma^2}{Nh}
\end{aligned}
\tag{17}
$$

Combining the bias and variance part, we have

$$
MSE_{hist}^{reg}(s) = [Bias_{hist}^{reg}(s)]^2 + Var_{hist}^{reg}(s) \leq L^2 h^2 + \frac{\sigma^2}{Nh},
\tag{18}
$$

It is easy to obtain the optimal bandwidth $h^* = [\frac{\sigma^2}{2L^2N}]^{\frac{1}{3}} = O(N^{-\frac{1}{3}})$, and the minimized MSE is lower than $[\frac{L\sigma^2}{2N}]^{\frac{2}{3}} = O(N^{-\frac{2}{3}})$. $\qquad\square$

### D.3 PROOF OF LEMMA 3

*Proof.* Note that there exists bias-variance tradeoff for $MSE_{kernel}^{pdf}(s)$, i.e.,

$$
\begin{aligned}
MSE_{kernel}^{pdf}(s) &= \mathbb{E}[|\hat{p}_S^h(s) - p_S(s)|^2] \\
&= \underbrace{\Big|\mathbb{E}[\hat{p}_S^h(s)] - p_S(s)\Big|^2}_{Bias_{kernel}^{pdf}(s)} + \underbrace{\mathbb{E}\Big[\Big|\hat{p}_S^h(s) - \mathbb{E}[\hat{p}_S^h(s)]\Big|^2\Big]}_{Var_{kernel}^{pdf}(s)},
\end{aligned}
$$

Next we analyze the bias and variance for probability density function MSE. For the bias part, the expectation of estimated probability density function satisfies:

$$
\begin{aligned}
\mathbb{E}[\tilde{p}_S^h(s)] - \tilde{p}_S^h(s) &= \mathbb{E}[\frac{1}{Nh}\sum_{n=1}^{N} K(\frac{s_n - s}{h})] - \tilde{p}_S^h(s) \\
&= \frac{1}{h}\mathbb{E}[K(\frac{s_n - s}{h})] - \tilde{p}_S^h(s) \\
&= \frac{1}{h}\int K(\frac{s_n - s}{h})p_S(s_n)\mathrm{d}s_n - \tilde{p}_S^h(s) \\
&= \int K(y)p_S(s + hy)\mathrm{d}y - \tilde{p}_S^h(s).
\end{aligned}
\tag{19}
$$

where the last equality holds by adopting transformation $y = \frac{s_n - s}{h}$. By Taylor expansion, when $h$ is small, we have

$$
p_S(s + hy) = p_S(s) + hyp_S'(s) + \frac{h^2 y^2}{2}p_S''(s) + o(h^2)
\tag{20}
$$

Based ob Eqs. (19) and (20), we have

$$
\begin{aligned}
Bias_{kernel}^{pdf}(s) &= |\mathbb{E}[\tilde{p}_S^h(s)] - \tilde{p}_S^h(s)| \\
&= p_S(s)\int K(y)\mathrm{d}y + hp_S'(s)\int yK(y)\mathrm{d}y + \frac{h^2}{2}p_S''(s)\int y^2 K(y)\mathrm{d}y + o(h^2) - p_S(s) \\
&= \frac{h^2\sigma_K^2}{2}p_S''(s);
\end{aligned}
\tag{21}
$$

As for the variance part, we have

$$
\begin{aligned}
Var_{kernel}^{pdf}(s) &= \mathbb{D}[\frac{1}{Nh}\sum_{n=1}^{N} K(\frac{s_n - s}{h})] = \frac{1}{Nh^2}\mathbb{D}[K(\frac{s_n - s}{h})] \leq \frac{1}{Nh^2}\mathbb{E}[K^2(\frac{s_n - s}{h})] \\
&= \frac{1}{Nh^2}\int K^2(\frac{s_n - s}{h})p_S(s_n)\mathrm{d}s_n = \frac{1}{Nh}\int K^2(y)p_S(s + hy)\mathrm{d}y \\
&= \frac{1}{Nh}\int K^2(y)[p_S(s) + hyp_S'(s) + o(h)]\mathrm{d}y = \frac{p_S(s)}{Nh}\int K^2(y)\mathrm{d}y + o(h) \\
&= \frac{p_S(s)\tilde{\sigma}_K^2}{Nh} + o(\frac{1}{N})
\end{aligned}
\tag{22}
$$

Combining the bias and variance part, we have

$$
\begin{aligned}
MSE_{kernel}^{pdf}(s) &= [Bias_{kernel}^{pdf}(s)]^2 + Var_{kernel}^{pdf}(s) \\
&\leq \frac{h^4|p_S''(s)|^2\sigma^4}{4} + \frac{p_S(s)\tilde{\sigma}_K^2}{Nh} + o(h^4) + o(\frac{1}{N}) \\
&= O(h^4) + O(\frac{1}{Nh}),
\end{aligned}
\tag{23}
$$

It is easy to obtain that the optimal bandwidth $h^* = O(N^{-\frac{1}{5}})$, and thus, the minimized MSE is lower than $O(N^{-\frac{4}{5}})$. □

### D.4 PROOF OF LEMMA 4

*Proof.* Recall that the mean prediction conditioned on sensitive attribute $s$ is given by $m(s) = \mathbb{E}[\hat{Y}|S = s]$, we rewrite prediction as $\hat{Y} = m(S) + e$, where $e$ is the regression noise and satisfies $\mathbb{E}[e] = 0$ and $E[e^2|S = s] = \sigma^2(s)$. Note that the prediction $Y = m(s) + [m(S) - m(s)] + e$, we have:

$$
\begin{aligned}
\frac{1}{Nh}\sum_{n=1}^{N}K(\frac{s_n - s}{h})\hat{y}_n &= \frac{1}{Nh}\sum_{n=1}^{N}K(\frac{s_n - s}{h})m(s) + \frac{1}{Nh}\sum_{n=1}^{N}K(\frac{s_n - s}{h})[m(s_n) - m(s)] \\
&\quad + \frac{1}{Nh}\sum_{n=1}^{N}K(\frac{s_n - s}{h})e_n \\
&= \hat{p}_S(s)m(s) + \tilde{m}_1(s) + \tilde{m}_2(s).
\end{aligned}
$$

Based on Eq. (13), we have

$$
\tilde{m}(s) = m(s) + \frac{\tilde{m}_1(s)}{\hat{p}_S(s)} + \frac{\tilde{m}_2(s)}{\hat{p}_S(s)}.
\tag{24}
$$

Since $\mathbb{E}[e|S = s] = 0$, we have the expectation of $\mathbb{E}[\tilde{m}_2(s)] = 0$ since $\mathbb{E}[K(\frac{s_n-s}{h})e] = \mathbb{E}[K(\frac{s_n-s}{h})\mathbb{E}[e|S = s_n]] = 0$. As for the variance of $\tilde{m}_2(s)$, we have

$$
\begin{aligned}
\mathbb{D}[\tilde{m}_2(s)] &= \frac{1}{Nh^2}\mathbb{E}[K(\frac{s_n - s}{h})e^2] = \frac{1}{Nh^2}\mathbb{E}[K(\frac{s_n - s}{h})\sigma^2(s_n)] \\
&= \frac{1}{Nh^2}\int K(\frac{s_n - s}{h})\sigma^2(s_n)p_S(s_n)\mathrm{d}s_n \\
&= \frac{1}{Nh}\int K(y)\sigma^2(s + yh)p_S(s + yh)\mathrm{d}s_n \\
&= \frac{\tilde{\sigma}_K^2\sigma^2(s)p_S(s)}{Nh} + o(\frac{1}{Nh});
\end{aligned}
\tag{25}
$$

Subsequently, we consider the expectation and variance of $\tilde{m}_1(s)$. Specifically, for expectation, we have

$$
\begin{aligned}
\mathbb{E}[\tilde{m}_1(s)] &= \frac{1}{h}\mathbb{E}\Big[K(\frac{s_n - s}{h})\Big(m(s_n) - m(s)\Big)\Big] \\
&= \frac{1}{h}\int K(\frac{s_n - s}{h})\Big(m(s_n) - m(s)\Big)p_S(s_n)\mathrm{d}s_n \\
&= \int K(y)\Big(m(s + hy) - m(s)\Big)p_S(s + hy)\mathrm{d}y \\
&= \int K(y)\Big(hym'(s) + \frac{y^2h^2}{2}m''(s)\Big)\Big(p_S(s) + hyp_S'(s)\Big) + o(h^2)\mathrm{d}y \\
&= \sigma_K^2h^2\Big(\frac{m''(s)p_S(s)}{2} + m'(s)p_S'(s)\Big) + o(h^2).
\end{aligned}
\tag{26}
$$

As for the variance of $\tilde{m}_1(s)$, we have

$$
\begin{aligned}
\mathbb{D}[\tilde{m}_1(s)] &= \frac{1}{Nh^2}\mathbb{D}\Big[K(\frac{s_n-s}{h})\big(m(s_n)-m(s)\big)\Big] \\
&= \frac{1}{Nh^2}\int\Big\{K(\frac{s_n-s}{h})\big(m(s_n)-m(s)\big)-\mathbb{E}[\tilde{m}(s)]\Big\}^2 p_S(s_n)\mathrm{d}s_n \\
&= \frac{1}{Nh}\int\Big[K(y)hym'(s)-\mathbb{E}[\tilde{m}(s)]\Big]^2\big(p_S(s)+yhp_S'(s)\big)\mathrm{d}s_n \\
&= \frac{\sigma^2(s)\sigma_K^2}{P_S(s)Nh}+o(\frac{1}{Nh})=O(\frac{1}{Nh}).
\end{aligned}
$$

Combining the bias and variance part, we have

$$
MSE_{kernel}^{reg}(s) = [Bias_{kernel}^{reg}(s)]^2 + Var_{kernel}^{reg}(s) \leq L^2h^2 + \frac{\sigma^2}{Nh} \leq O(h^4)+O(\frac{1}{Nh}), \quad (27)
$$

Based on the inequality of arithmetic and geometric means, it is easy to obtain the optimal bandwidth $h^* = O(N^{-\frac{1}{5}})$, and the minimized MSE is lower than $O(N^{-\frac{4}{5}})$. $\qquad\square$

## E    PROOF OF THEOREM 4

Given predictor $\hat{Y} = f(X)$, adversary $\hat{S} = g(\hat{Y})$, and adversary utility $\mathcal{L}_{adv}(\hat{S},S) = 1-|\hat{S}-S|$, GDP and adversary utility are given by

$$
\Delta GDP = \mathbb{E}_S\Big[\big|\mathbb{E}_{X|S}[f(X)]-\mathbb{E}_X[f(X)]\big|\Big]; \qquad \mathcal{L}_{adv} = \mathbb{E}_S\Big[\mathbb{E}_{X|S}\big[1-|S-g(f(X))|\big]\Big].
$$

Intuitively, higher model prediction implies larger sensitive attribute if mean prediction function $m(s)$ is more close to $s$ compared with $1-s$ and vice versa. Therefore, we construct adversary $g$ as follows:

$$
g^{\#}\big(f(X)\big) = \begin{cases} f(X), & \text{if } \mathbb{E}_S[|S-m(S)|]\leq\mathbb{E}_S\big[\big|S-\big(1-m(S)\big)\big|\big]; \\ 1-f(X), & \text{Otherwise.} \end{cases}
$$

Suppose without loss of generality (WLOG) that $\mathbb{E}_S[|S-m(S)|]\leq\mathbb{E}_S\big[\big|S-\big(1-m(S)\big)\big|\big]$, i.e., higher model prediction implies larger sensitive attribute. Then adversary utility is given by

$$
\begin{aligned}
\mathcal{L}_{adv}\Big(g^{\#}\big(f(X)\big),S\Big) &= \mathbb{E}_S\Big[\mathbb{E}_{X|S}\big[1-|S-f(X)|\big]\Big] \leq \mathbb{E}_S\Big[\big[1-|S-\mathbb{E}_{X|S}[f(X)]|\big]\Big] \\
&= \mathbb{E}_S\Big[1-|S-m(S)|\Big]
\end{aligned}
$$

where the inequality holds due to Jensen's inequality and convex function $|x-t|$ for any constant $t$. Next we show, under the constructive adversary $g$, the adversarial utility $\mathcal{L}_{adv}\geq\frac{1}{2}\geq\Delta GDP$. Firstly, notice that, for any function $m(S)\in[0,1]$ and $S\in[0,1]$, we have

$$
\begin{aligned}
|S-m(S)|+\big|S-\big(1-m(S)\big)\big| \leq \max\Big\{&|1-m(S)|+\big|1-\big(1-m(S)\big)\big|, \\ &|0-m(S)|+\big|0-\big(1-m(S)\big)\big|\Big\}=1,
\end{aligned}
$$

which implies that

$$
\begin{aligned}
\mathbb{E}_S\Big[1-|S-m(S)|\Big] &\leq \frac{1}{2}\Big(\mathbb{E}_S[|S-m(S)|]+\mathbb{E}_S\big[\big|S-\big(1-m(S)\big)\big|\big]\Big) \\
&= \frac{1}{2}\Big(\mathbb{E}_S\big[|S-m(S)|+\big|S-\big(1-m(S)\big)\big|\big]\Big)\leq\frac{1}{2}.
\end{aligned}
$$

As for the analysis on GDP, we consider the worst case of mean prediction function $m(s)$ since GDP satisfies $\Delta GDP = \mathbb{E}_S\big[\big|m(S)-\mathbb{E}_S[m(S)]\big|\big]$. Note that function $|x|$ is strictly convex and the solution to maximize a strictly convex function over all finite support given first moment is achieved by a distribution of two mass extreme points, GDP can achieve maximal value when $m(S)$ is 0 or 1. Define $p_{pos} = \mathbb{E}_S\Big[P\big(m(S)=1\big)\Big]$, then $\mathbb{E}_S[m(S)] = p_{pos}$ and $\Delta GDP = p_{pos}\big(1-p_{pos}\big)+(1-p_{pos})p_{pos}\leq\frac{1}{2}$. Therefore, for the optimal adversary $g^*$, we have

$$
\mathcal{L}_{adv}\Big(g^*\big(f(X)\big),S\Big)\geq\mathcal{L}_{adv}\Big(g^{\#}\big(f(X)\big),S\Big)\geq\Delta GDP. \tag{28}
$$

## F  DATA STATISTICS

For fair comparision with previous work, we perform the classification and regression task on five datasets, including Crimes, Adult, Pokec-n, Pokec-z and Harris dataset. The first four dataset have been widely adopted to study the fairness problem in tabular data and graph data, while Harris dataset is collected by ourself for temporal graph data. Table 1 presents additional information on the real-world tabular, graph and temporal graph datasets. For task type column, "Reg" and "Clf" represents regression task and classification task, respectively.

Table 1: Statistical Information on Datasets

| Data Type | Dataset | Task Type | # Nodes /Samples | # Edges | # Features | Metric | |
|---|---|---|---|---|---|---|---|
| Tabular | Crimes | Reg | 1994 | — | 121 | MAE | |
| | Adult | Clf | 45222 | — | 13 | | |
| Graph | Pokec-n | Clf | 66569 | 729129 | 59 | | GDP |
| | Pokec-z | Clf | 67797 | 882765 | 59 | Acc | |
| Temporal Graph | Harris | Clf | 4204 | 19946 | 36 | | |

## G  MORE DETAILS ON SYNTHETIC EXPERIMENTS

### G.1  GDP CALCULATION

We firstly provide ground truth analysis in synthetic experiments so that we can evaluate propsoed two GDP estimation methods error. Considering bivariate Gaussian distribution with mean $\mu = [\mu_1, \mu_2]$ and covariance matrix $\Sigma = \begin{bmatrix} \sigma_{11} & \sigma_{12} \\ \sigma_{21} & \sigma_{22} \end{bmatrix}$ and note that covariance matrix is positive definite matrix, it is easy to obtain inverse covariance matrix $\Sigma^{-1} = \begin{bmatrix} \lambda_{11} & \lambda_{12} \\ \lambda_{21} & \lambda_{22} \end{bmatrix}$, where $\lambda_{22} = \frac{\sigma_{11}}{|\Sigma|}$ and $\lambda_{12} = -\frac{\sigma_{12}}{|\Sigma|}$. The joint distribution of $(S, \hat{Y})$ follows $p_{S,\hat{Y}}(s,\hat{y}) = \frac{1}{\sqrt{2\pi|\Sigma|}} \exp\left( -\frac{1}{2}\lambda_{11}(s-\mu_1)^2 - \frac{1}{2}\lambda_{22}(s-\mu_2)^2 + \lambda_{12}(s-\mu_1)(\hat{y}-\mu_2) \right)$. Based on probability theory, we can have the condition function $p_{\hat{Y}|S}(\hat{y}|s)$ as follows:

$$
\begin{aligned}
p_{\hat{Y}|S}(\hat{y}|s) &= \frac{p_{S,\hat{Y}}(s,\hat{y})}{p_S(s)} = \frac{1}{\sqrt{\frac{2\pi}{\lambda_{22}}}} \exp\left( -\frac{\lambda_{22}\left(\hat{y} - \frac{\lambda_{22}\mu_2 + \lambda_{12}s - \lambda_{12}\mu_1}{\lambda_{22}}\right)^2}{2} \right) \\
&\sim \mathcal{N}\left( \frac{\sigma_{11}\mu_2 + \sigma_{12}(s-\mu_1)}{\sigma_{11}}, \frac{|\Sigma|}{\sigma_{11}} \right).
\end{aligned}
$$

which means the mean prediction function $m(s) = \frac{\sigma_{11}\mu_2 + \sigma_{12}(s-\mu_1)}{\sigma_{11}}$. Notice that the probability density function of sensitive attribute is also Guassian with $\mathcal{N}(\mu_1, \sigma_{11})$, therefore, the GDP is

$$
\begin{aligned}
\Delta GDP &= \int |m(s) - \mu_2| p_S(s) \mathrm{d}s \\
&= \int \left| \frac{\sigma_{12}(s-\mu_1)}{\sigma_{11}} \right| \frac{1}{\sqrt{2\pi\sigma_{11}}} \exp\left( -\frac{(s-\mu_1)^2}{2\sigma_{11}} \right) \mathrm{d}s = \frac{2\sigma_{12}}{\sqrt{2\pi\sigma_{11}}}.
\end{aligned}
$$

Next, we calculate GDP for the second synthetic probability density function $p_{S,\hat{Y}}(s,\hat{y}) = s + \hat{y}$ if $0 \le s, \hat{y} \le 1$. It is easy to obtain the conditional probability $p_{\hat{Y}|S}(\hat{y}|s) = \frac{s+\hat{y}}{s+\frac{1}{2}}$ if $0 \le s, \hat{y} \le 1$ and thus the mean prediction function $m(s) = \mathbb{E}[\hat{Y}|S=s] = \frac{\frac{1}{2}s + \frac{1}{3}}{s+\frac{1}{2}}$. Similarly, the probability of

sensitive attribute is $p_S(s) = s + \frac{1}{2}$ if $0 \le s, \hat{y} \le 1$. Thus, the GDP satisfies

$$\Delta GDP = \int \left| m(s) - \mathbb{E}[m(S)] \right| p_S(s) \mathrm{d}s = \int_0^1 \left| \frac{\frac{1}{2}s + \frac{1}{3}}{s + \frac{1}{2}} - \frac{7}{12} \right| (s + \frac{1}{2}) \mathrm{d}s = \frac{1}{48}.$$

### G.2 MORE SYNTHETIC EXPERIMENT RESULTS ON ESTIMATION ERROR

Figure 6 shows local prediction average and sensitive attribute probability density function estimation results for different kernel function choice. The top two subfigures show the local prediction average estimation error w.r.t. bandwidth choice. It is seen that the tricube and aitchison aitken kernel function achieve better and robust local prediction average estimation compared with Gaussian kernel function. The mid subfigures show the local prediction average result for different sensitive attribute and bottom two subfigures shows probability density function estimation results with different kernel and histogram choice. It is seen that kernel estimation possesses more smooth and accurate probability density function estimation.

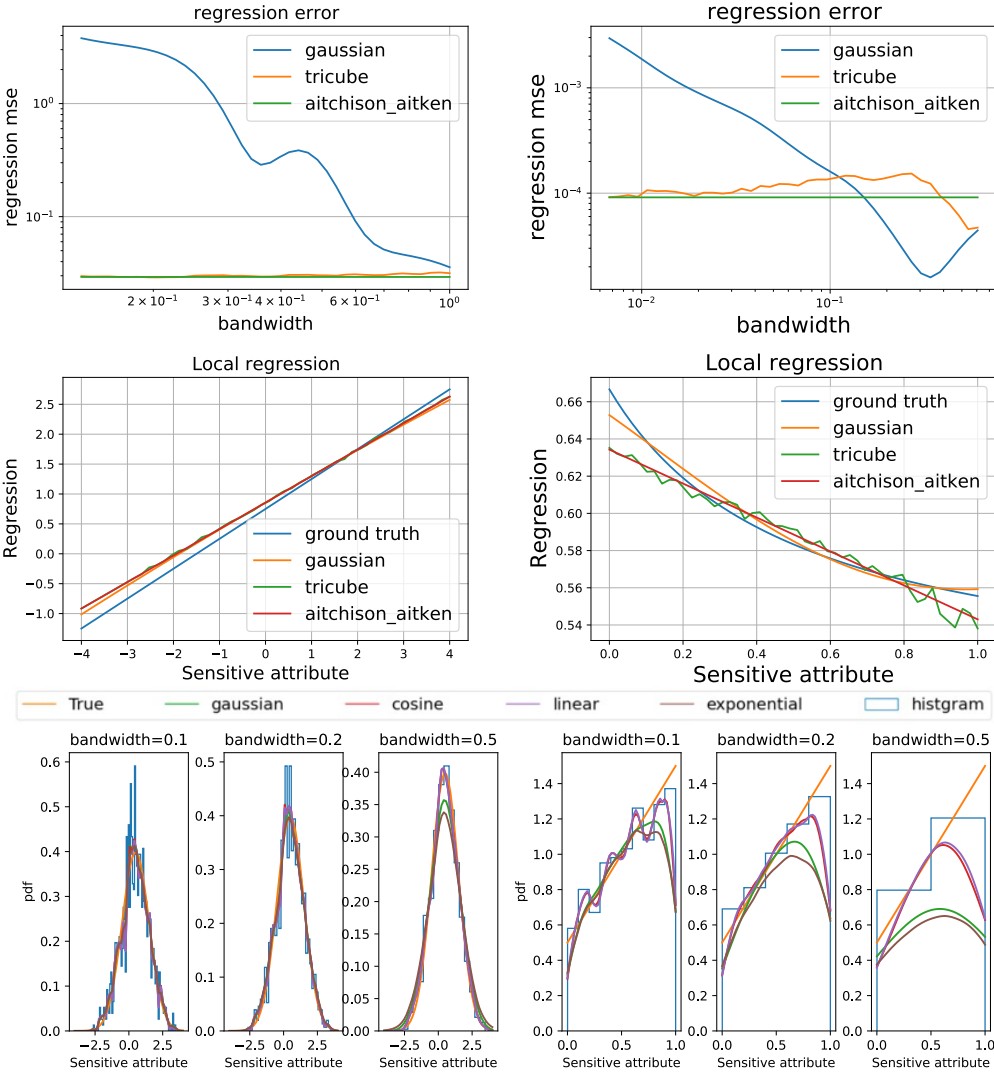

Figure 6: Local prediction average and sensitive attribute probability density function estimation error analysis with respect to the kernel bandwidth and number of samples for bivariate Gaussian distribution and second synthetic distributions.

### G.3 MORE SYNTHETIC EXPERIMENT RESULTS ON ROBUSTNESS

In this subsection, we aim to investigate the robustness of GDP estimation error over different mean and covariance parameters. Note that only the parameters expectation $\mu = [\mu_1, \mu_2] = [0.5, 1.0]$ and covariance matrix $\Sigma = \begin{bmatrix} \sigma_{11} & \sigma_{12} \\ \sigma_{12} & \sigma_{22} \end{bmatrix} = \begin{bmatrix} 1 & 0.5 \\ 0.5 & 2.0 \end{bmatrix}$ are adopted on synthetic experiments, we adopt these parameters as default and only modify the one parameter to inspect GDP estimation error. Figures 7 and 8 show the GDP estimation error with mean parameters $\mu_1$ and $\mu_2$, and covariance parameters $\sigma_{11}$, $\sigma_{12}$, and $\sigma_{22}$ given the same data size 1000. We have two observations: (1) kernel-based GDP estimation methods consistently achieves lower estimation errors than histogram-based counterpart for any Gaussian distribution parameter, which further validates the superiority of kernel-based method. (2) kernel-based GDP estimation error is almost the same for any kernel function pair choice, which validates our theoretical analysis in Theorem. 3.

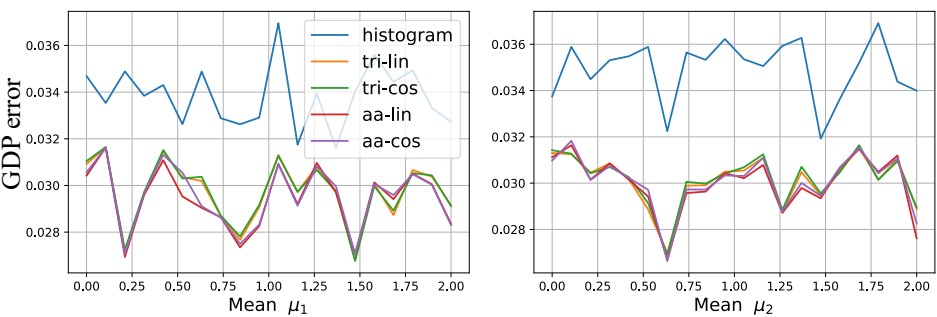

Figure 7: GDP estimation error for different mean parameters

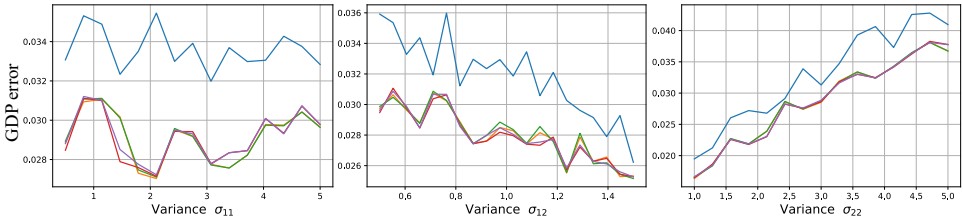

Figure 8: GDP estimation error for different covariance parameters

## H MORE DETAILS ON REAL-WORLD EXPERIMENTAL RESULTS

Many real-world data are organized as tabular, graph and temporal graph data. Various machine learning models and training strategy are specifically developed for these data types (Chen & Guestrin, 2016; Han et al., 2022; Jiang et al., 2022; Zhao et al., 2019). Our experiments demonstrate that the proposed GDP and corresponding regularizer can be adopted for these data types.

### H.1 MITIGATION PERFORMANCE FOR TEMPORAL GRAPH DATA

The temporal graph data, provided by data intelligence company Cuebiq (Cuebiq, 2021), is collected from anonymous human movement activities, including the coordinates and time of mobile devices at stop points, during August 2017 in Harris County (Houston) Texas, USA. To generate the temporal graph, we first divide the Harris County into several grid cells with equal size approximately $1km \times 1km$. Each grid cell is treated as a graph node, and temporal link between two nodes represents at least one user movement in hourly basis duration. The node features are generated from socio-demographic data of the American Community Survey (ACS) 2014–2018 (5-year) data by the U.S. Census Bureau (Bureau, 2021). In the experiment, the white race ratio is treated as continuous sensitive attributes and our task is to predict whether the income of each node is high

or low. We adopt temporal graph attention (TGAT) (da Xu et al., 2020)[8] with map and product attention mechanism to efficiently aggregate temporal-topological neighborhood features and report the mean prediction performance and GDP with 5 running times.

We compare the mitigation performance of kernel estimation and other baselines for private Harris dataset with two backbones in Figure 9. Simialrly, the hyper-parameter $\lambda$ control the tradeoff between accuracy and GDP. Again, kernel estimation consistently outperforms the other baselines by a large margin and binary-quantized sensitive attributes inevitably deteriorate the mitigation performance.

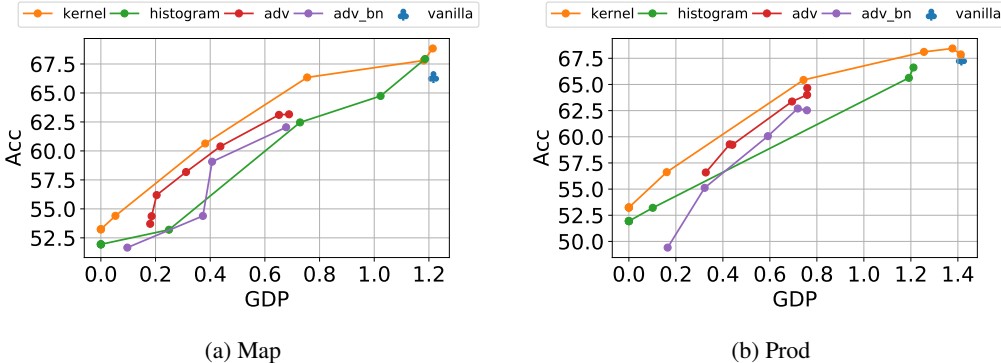

(a) Map        (b) Prod

Figure 9: Mitigation performance for temporal graph Harris dataset. (a) TGAT with map attention; (b) TGAT with product attention.

## H.2 PREDICTION PERFORMANCE AND GDP TRADEOFF CURVE DURING TRAINING

**Training curve on tabular data**: Aiming to inspect the dynamic prediction performance and GDP during model training, we provide prediction performance and GDP tradeoff curve for Adult and Crimes dataset in Figure 10. The left and right y-axis represent the prediction performance and GDP metric, respectively. It is seen that, for kernel or histogram as regularization, the hyperparameter can control the prediction performance and GDP tradeoff, while the training is highly unstable with large variance for adversarial debiasing. Additionally, kernel estimation as regularization possesses better bias mitigation performance for Adult and Crimes dataset.

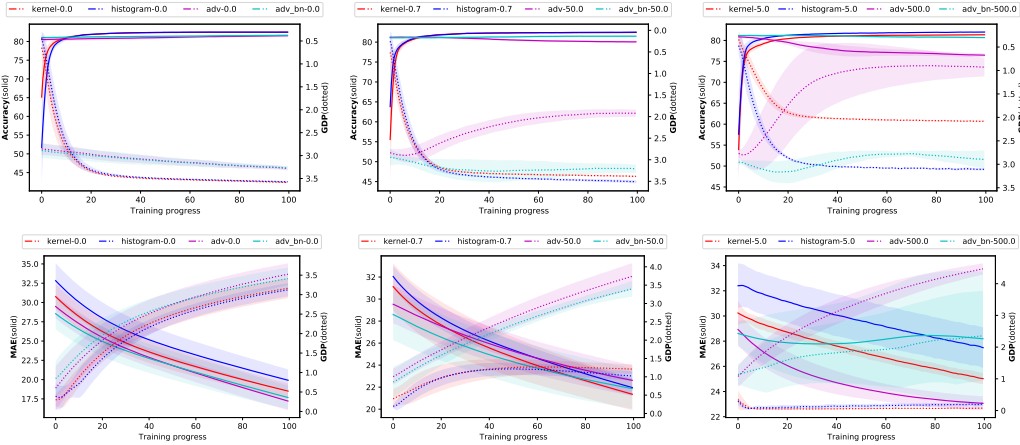

Figure 10: Prediction performance and GDP training curve for Adult (top) and Crimes (bottom) datasets with different hyperparameters.

**Training curve on graph data**: Figures 11 and 12 demonstrates prediction performance and GDP tradeoff curve for Pokec-n and Pokec-z datasets using GCN and GAT model. It is seen that, for

---

[8]https://github.com/StatsDLMathsRecomSys/Inductive-representation-learning-on-temporal-graphs

kernel or histogram as regularization, the hyperparameter can control the prediction performance and GDP tradeoff, while the training is highly unstable with large variance for adversarial debiasing. Additionally, kernel estimation as regularization possesses better bias mitigation performance for Pokec-n and Pokec-z datasets in GCN and GAT model.

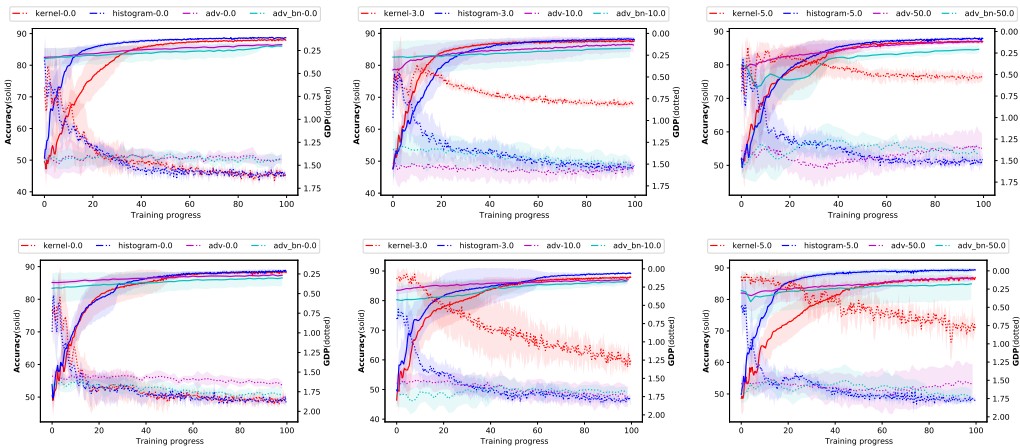

Figure 11: Prediction performance and GDP training curve for Pokec-n (top) and Pokec-z (bottom) datasets with GAT model.

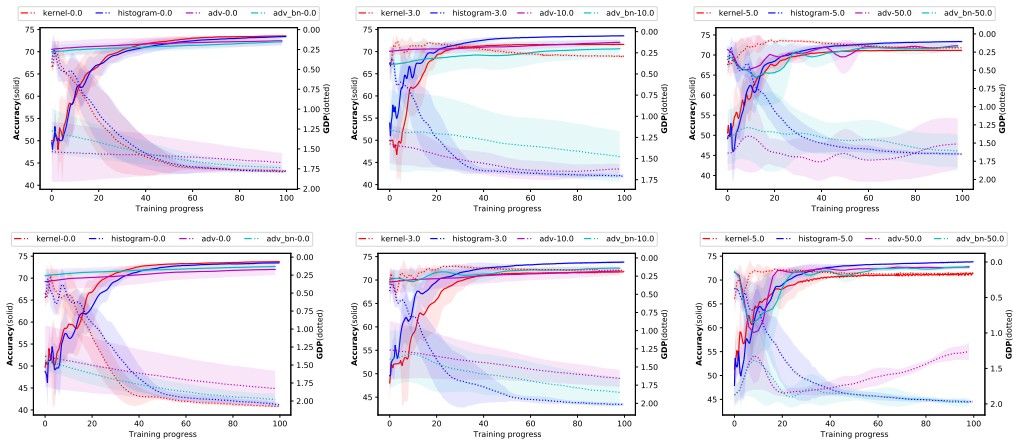

Figure 12: Prediction performance and GDP training curve for Pokec-n (top) and Pokec-z (bottom) datasets with GCN model.

**Training curve on temporal graph data**: Figure 13 demonstrates prediction performance and GDP tradeoff curve for Harris datasets using TGAT with map and product attention mechanism. It is seen that, for kernel or histogram as regularization, the hyperparameter can control the prediction performance and GDP tradeoff, while the training is highly unstable with large variance for adversarial debiasing. Additionally, kernel estimation as regularization possesses better bias mitigation performance for TGAT with map and product attention mechanism.

**Training curve on compositional sensitive attribute**: Figure 14 demonstrates prediction performance and GDP tradeoff curve for Adult and Crimes dataset. It is seen that, for kernel or histogram as regularization, the hyperparameter can control the prediction performance and GDP tradeoff, while the training is highly unstable with large variance for adversarial debiasing. Additionally, kernel estimation as regularization possesses better bias mitigation performance for Adult and Crimes dataset.

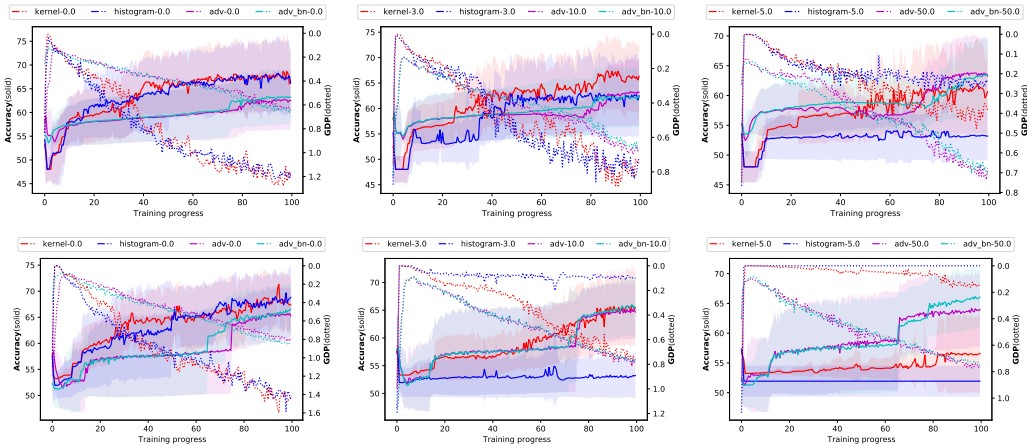

Figure 13: Prediction performance and GDP training curve for TGAT with map (top) and product (bottom) attention mechanism with Harris data.

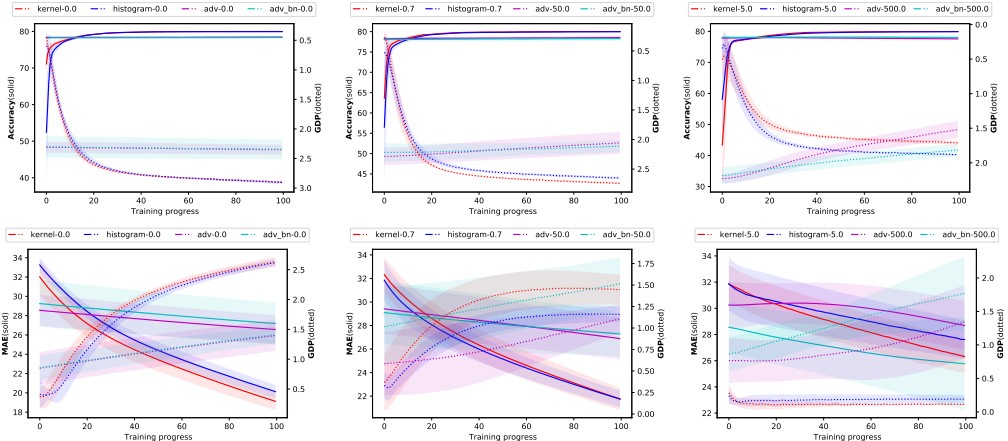

Figure 14: Prediction performance and GDP training curve for Adult (top) and Crimes (bottom) datasets with compositional attributes.

# I  FUTURE WORKS

There are several interesting future work related to proposed GDP:(1)Extend other fairness metrics, such as Equal Odds (EO), for continuous and discrete sensitive attributes. The connection between extended fairness metric and adversarial debiasing is also interesting. (2) Provide a more fine-grained analysis on estimation error, including the exact optimal bandwidth and estimation error expression, the lower bound of the estimation error for histogram and kernel GDP estimation over all data distribution and model prediction, the estimation error analysis over compositional sensitive attributes. (3)Investigate the better GDP estimation method with faster convergence guarantee.

