# OpenReview forum: "Generalized Demographic Parity for Group Fairness"
_ICLR.cc/2022/Conference — ICLR 2022 Poster_

### Official Review · Reviewer_dM4B · 2021-10-28

**Correctness:** 4
**Technical Novelty And Significance:** 2
**Empirical Novelty And Significance:** 2
**Recommendation:** 5
**Confidence:** 4

**Main Review:**

Overall, the paper is well-written and easy to follow. I like the structure of this paper: it starts with a new definition for measuring fairness; introduces ways to estimate this fairness measure; and presents applications of this measure. However, my main concern is the technical novelty of this paper (see my detailed comments listed below).

1. My main concern is the estimation error bound provided in Theorem 3. It seems that this result relies on a strong assumption (i.e., the prediction function is Lipschitz continuous w.r.t. the sensitive attribute) and it is unclear to me the order presented in this theorem is optimal.

(1) [Strong assumption.] It is very hard for me to come up with a scenario where the Lipschitz condition is satisfied and the Lipschitz constant is small. Note that the Lipschitz constant will affect the optimal bandwidth choice and, potentially, the sample complexity. It would be great if the authors can provide an example to show that this Lipschitz condition is inevitable.

(2) [Optimal order.] The order of estimation error given in Theorem 3 is O(N^{-4/5}). However, it is unclear to me if this is the optimal order. Can the authors derive some converse results to prove that this order cannot be improved anymore?

2. [Intersectionality.] The authors only consider a single sensitive attribute. What if there are multiple sensitive attributes (e.g., age, weight, income, …). How does the number of sensitive attributes affect the estimation error bound in Theorem 3? Does the bound suffer from the curse of dimensionality in this case?

3. The authors wrote in Theorem 3 “Under the optimal bandwidth choice, …”. Can the authors specify what the optimal bandwidth is?

4. Can the authors include an error bar for each figure in the paper?

**Summary Of The Paper:**

This paper proposes a new extension of demographic parity when the sensitive attribute is continuous. It proposes two techniques to estimate this quantity from data. The authors also consider adding this quantity as a regularization term to control discrimination while designing machine learning models.

**Summary Of The Review:**

The paper is interesting but the technical novelty may not be sufficient

---

> ### Author Response · Authors · 2021-11-19
> **Response to Reviewer dM4B**
>
> We thank the reviewer for the constructive comments and appreciate the structure of this manuscript.
>
> **Q1: The estimation error bound provided in Theorem 3 relies on a strong assumption (i.e., the prediction function is Lipschitz continuous w.r.t. the sensitive attribute). Note that the Lipschitz constant will affect the optimal bandwidth choice and, potentially, the sample complexity. It would be great if the authors can provide an example to show that this Lipschitz condition is inevitable.**
>
> A1: Thanks for your insightful comment. We argue that Lipschitz continuous assumption is **not a strong assumption** since we do **not necessarily require small Lipschitz constant**. The Lipschitz continuous assumption only requires the existence of Lipschitz constant, i.e., **bounded gradient** w.r.t. sensitive attribute is sufficient for our assumption. Therefore, we believe that such an assumption is easily satisfied in practice.
>
> As for the influence of Lipschitz constant, we totally agree with the reviewer that Lipschitz constant will affect the optimal bandwidth and estimation error. However, we provably show **negligible influence** since **Lipschitz constant does not influence the order of the optimal bandwidth and GDP estimation error.** In other words, Lipschitz constant has a quite limited influence on bandwidth choice and GDP estimation error.
>
> As for the inevitability of Lipschitz continuous assumption, we can provide a very intuitive justification: for local average estimation, it is impossible to obtain a reasonable estimation based on **neighbor data samples** without Lipschitz continuous assumption since the behavior of neighbor data samples could be quite different. The Lipschitz continuous assumption support that the local average can be estimated based on neighbor data samples. We have added the justification on the footnote of page 5.
>
> **Q2: [Optimal order.] The order of estimation error given in Theorem 3 is $O(N^{-\frac{4}{5}})$. However, it is unclear to me if this is the optimal order. Can the authors derive some converse results to prove that this order cannot be improved anymore?**
>
> A2: Thanks for your insightful comment. We provably show that the GDP estimation error can be upper bounded by $O(N^{-\frac{4}{5}})$,  but we are not sure whether the order is optimal or not. The positive lower bound **does not exist** under Lipchitz continuous assumption.
>
> **On the superiority of kernel-based estimation method:** Although we do not know whether the convergence rate is optimal or not, Theorem 3 can still provide insights for GDP estimation methods. First, kernel-based and histogram-based estimation methods both possess the **theoretical foundation toward accurate estimation** as long as the data size is sufficiently large. Second, Theorem. 3 justifies the superiority of kernel-based estimation method with faster provably convergence rate. We also validate the faster convergence rate via synthetic experiments.
>
> **On the Lower bound:** For the general case, the actual estimation error could be zero. For example, if the model always provides the constant prediction for any input (i.e., zero Lispchitz constant), then the estimated GDP must be 0 since the local average is always equivalent to the global average. In this case, the estimation error is 0 for any data size. Therefore, for the converse part, it is impossible to provide the lower bound for the general case.
>
> **Q3: [Intersectionality.] IF there are multiple sensitive attributes (e.g., age, weight, income, …), how does the number of sensitive attributes affect the estimation error bound in Theorem 3? Does the bound suffer from the curse of dimensionality in this case?**
>
> A3: Thanks for raising this interesting point. We totally agree that
> the convergence rate of estimation error for multiple sensitive attributes would be slower since GDP estimation would be more challenging given the same data size, but is out of the scope of this paper. Hence, we have provided some discussion on the estimation error analysis for multiple sensitive attributes
>
> As for the curse of dimensionality for GDP estimation, the number of continuous sensitive attributes, such as age, weight, and income, is quite limited in real-world datasets. Generally speaking, the number of continuous sensitive attributes is usually less than 5 due to the limited access of the sensitive attributes by law. Hence, we argue that the estimation error does not suffer from the curse of dimensionality in practice.
>
>
> **Q4: The authors wrote in Theorem 3 ``Under the optimal bandwidth choice, …”. Can the authors specify what the optimal bandwidth is?**
>
> A4: Thanks for this constructive comment. In Theorem 3, we have specified the optimal bandwidth as $h_{hist}^{\*}=O(N^{-\frac{1}{3}})$ and $h_{kernel}^{\*}=O(N^{-\frac{1}{5}})$. We have also added the error bar for Figures 3-5 in the revised version.

---

> ### Author Response · Authors · 2021-11-21
> **Appreciate further comments**
>
> Dear Reviewer dM4B,
>
> We really appreciate your previous constructive comments and have addressed your concerns one by one. As the discussion period is approaching its end, we would really appreciate it if you could kindly let us know there are any further questions. We will be more than happy to address them fully.
>
> Yours Sincerely,
>
> Authors

---

> ### Comment · Reviewer_dM4B · 2021-11-22
> **Follow-up**
>
> Thank the authors for providing a thoughtful response! I have some follow-up questions based on the response.
> 1. [Order vs constant.] When considering estimation error, the authors mainly focus on the order bounds. However, I believe the precise expression of the estimation error bound is also important, especially when the number of data is limited. This is why I was concerned about the Lipschitz constant in my review. I agree with the authors that the Lipschitz constant does not affect the order of the estimation error. However, the Lipschitz constant may affect the constant, which is omitted from the order bound. In practice, this constant may dominate the estimation error for finite n. Similarly, for my Q4, I would like to see a precise expression of the optimal bandwidth instead of an order.
> 2. [Lower bound.] The counter-example given in the response is a bit weak. In the study of nonparametric estimation, people often formalize the estimation error as a min-max quantity. In the setting considered in this paper, if I understand correctly, min is over all estimators which maps n data points to a real number and max is over all data distributions and prediction models. Then the example given in the response could not give a clear lower bound for this min-max quantity because the max is over all models.
> 3. [Lipschitz constant.] Finally, I still think the Lipschitz constant assumption is strong. Note that it is equivalent to the notion of individual fairness. And training a classifier with individual fairness constraints is, in general, very challenging. It would be great if the authors can provide any practical examples or some algorithms in which this assumption is satisfied.

---

> > ### Author Response · Authors · 2021-11-22
> > **Response to Reviewer dM4B [part 1/2: order and lower bound]**
> >
> > Thanks for your constructive comments.
> >
> > Q1[Order vs constant.] When considering estimation error, the authors mainly focus on the order bounds. However, I believe the precise expression of the estimation error bound is also important, especially when the number of data is limited. This is why I was concerned about the Lipschitz constant in my review. I agree with the authors that the Lipschitz constant does not affect the order of the estimation error. However, the Lipschitz constant may affect the constant, which is omitted from the order bound. In practice, this constant may dominate the estimation error for finite n. Similarly, for my Q4, I would like to see a precise expression of the optimal bandwidth instead of an order.
> >
> > **A1:** Thanks for your constructive comments. We respectfully refuse to add a constant of the optimal bandwidth in Theorem 3. The reason is two-fold:
> >
> > - **From the technical perspective:** The exact analysis for optimal bandwidth or estimation error for finite data size is very challenging. Our current GDP estimation error analysis framework is referenced from nonparameter estimation and mainly relies on Taylor expansion. Therefore, the exact analysis is beyond the current analysis framework. Additionally, based on Lemmas 1 and 2, the constant of optimal bandwidth for local regression and sensitive attribute probability density function estimation is **different**, although the order of optimal bandwidth is the same. Therefore, we can only provide the order of optimal bandwidth and estimation error.
> >
> > - **From a real-world perspective:** In real-world datasets, the data sample size is usually sufficiently large for neural network training. In Appendix F, we provide the statistical information on real-world datasets and 4 dataset size largely exceed 2000, and only the size of Crimes dataset is 1994. Additionally, the synthetic experiments show that the GDP estimation error over the true GDP is less than $2\\%$ of the true GDP when data size is larger than $2000$. Therefore, we believe that kernel GDP estimation is sufficiently reliable to measure the fairness metric on real-world datasets.
> >
> > Furthermore, I would like to clarify that the goal of Theorem 3 is to **provide theoretical guarantee for histogram and kernel-based GDP estimation methods** and **provably shows the superiority of kernel-based GDP in the asymptotic sense**. Note that for mutual information estimation methods, such as Donsker-Varadhan representation [1], variational bounds [2], there is **no any theoretical guarantee on the estimation error**, we believe that the results of Theorem 3 have already demonstrated the superiority of GDP and validate the effectiveness of our proposed GDP estimation methods. I hope the reviewer can understand this situation.
> >
> > [1] Mohamed Ishmael Belghazi, Aristide Baratin, Sai Rajeshwar, Sherjil Ozair, Yoshua Bengio, Aaron
> > Courville, and Devon Hjelm. Mutual information neural estimation. In International Conference
> > on Machine Learning, pp. 531–540. PMLR, 2018.
> >
> > [2] Ben Poole, Sherjil Ozair, Aaron Van Den Oord, Alex Alemi, and George Tucker. On variational
> > bounds of mutual information. In International Conference on Machine Learning, pp. 5171–
> > 5180. PMLR, 2019.
> >
> > Q2[Lower bound.] The counter-example given in the response is a bit weak. In the study of nonparametric estimation, people often formalize the estimation error as a min-max quantity. In the setting considered in this paper, if I understand correctly, min is over all estimators which maps n data points to a real number and max is over all data distributions and prediction models. Then the example given in the response could not give a clear lower bound for this min-max quantity because the max is over all models.
> >
> > **A2:** Thanks for your insightful comment. We totally agree that the min-max quantity is a reasonable problem formulation for lower bound investigation and respectfully appreciate the reviewer for pointing out many valuable follow-up directions! I would like to remind the reviewer that Theorem 3 aims to theoretically justify the proposed histogram and kernel-based GDP estimation methods and provably shows the faster convergence rate of kernel-based GDP estimation methods. We believe that the current version of Theorem 3 has **already convincingly** demonstrated the **effectiveness** of kernel-based GDP estimation method. We also summarize several valuable directions in the Section Future works. Please see Appendix-I for more details.

---

> > ### Author Response · Authors · 2021-11-22
> > **Response to Reviewer dM4B [part 2/2: Lipschitz constant]**
> >
> > Q3[Lipschitz constant.] Finally, I still think the Lipschitz constant assumption is strong. Note that it is equivalent to the notion of individual fairness. And training a classifier with individual fairness constraints is, in general, very challenging. It would be great if the authors can provide any practical examples or some algorithms in which this assumption is satisfied.
> >
> > **A3:** I would like to emphasize that Lipschitz constant assumption **on local average** is **naturally satisfied** in practice.
> >
> > - Firstly, the neural network is usually designed as a continuous function w.r.t. the input features in real-world applications. Therefore, the local average (average model prediction given sensitive attribute) should be continuous w.r.t. sensitive attribute value.
> >
> > - Secondly, the well-trained neural network output is not sensitive to the small perturbation of sensitive attribute value due to **finite resolution of the weight and input**. In other words, the extremely small perturbation would not influence the neural network output.
> >
> > In a nutshell, the neural network model output is continuous and bounded, and not sensitive to small perturbation. Note that the local average is the average model prediction given sensitive attribute, Lipschitz constant assumption obviously holds for different fixed sensitive attributes. When sensitive attributes pair is infinitely close, the neural network output should keep the same.
> > Therefore, we think that Lipschitz constant assumption is reasonable.
> >
> > Compared with individual fairness, Lipschitz constant assumption on local average is **highly different** since this assumption requires that **local average** difference is bounded for **each sensitive attribute pair** while, for individual fairness, the model prediction difference should be bounded for **each sample pair**. I would like to say that Lipschitz constant assumption **on local average** is an easy-to-hold assumption compared with individual fairness.

---

### Official Review · Reviewer_Lp8E · 2021-11-02

**Correctness:** 4
**Technical Novelty And Significance:** 4
**Empirical Novelty And Significance:** 3
**Recommendation:** 6
**Confidence:** 3

**Main Review:**

Comments:
Section 1: Age is not a particularly successful example of a continuous attribute. A better example might be income differences (although it is not always legally protected, it is definitely of interest to decision-makers).

Section 3: The definition of GDP is precise and relatively natural. However, some justification of this particular choice is necessary. What are potential other definitions? Why is this definition the most natural (and technically sound) extension of the categorical definition? Explaining in simple terms would help provide intuition for the reader.

I think the theoretical foundations are a nice contribution of the paper. I would appreciate a proof sketch or intuition after each theorem.

Although I understand that this might be part of follow-up work, I think it is important to include a discussion on how other metrics can be similarly extended to capture continuous attributes. Demographic parity is a natural first step, though.

Section 6.1: why were these parameters chosen? Have robustness checks been performed and what were the results?


**Summary Of The Paper:**

The paper is motivated by the need to account for continuous attributes in fair Machine Learning. In particular, it proposes a Generalized Demographic Parity metric (GDP), i.e., a group fairness metric that can work with both continuous and discrete variables. A key challenge in doing so is preserving tractable computation.

Theoretically, GDP is defined via the weighted total variation distance and measures the distance between the local and global prediction average, where the weight corresponds to the pdf of the continuous sensitive attributes. Since the joint distribution between model prediction and sensitive attributes might not be available in practice, the paper also proposes two estimation methods: histogram estimation and kernel estimation. The former quantizes the continuous sensitive attributes into bins, whereas in the latter method the group indicator of the sample is treated as a kernel function. The kernel method leads to faster estimation error convergence rate in terms of sample size.
The paper provides an extensive evaluation with tabular, graph and temporal graph data, synthetic experiments, and classification and regression tasks. It is demonstrated experimentally that the kernel method has better bias mitigation performance.



**Summary Of The Review:**


In my opinion, the paper touches upon a very important and practical topic in fair ML. Not all attributes are categorical and privacy concerns might require fairness in a neighborhood or region of attributes. Thus, accounting for continuous variables in fairness definitions is crucial in many practical contexts.

The paper is methodologically sound. It provides a quite extensive theoretical justification and guarantees as well as a complete and convincing empirical evaluation with several datasets. It is well written and has the potential to lead to follow up work (both in terms of other definitions and also inspiring research on continuous attributes in fair ML).

---

> ### Author Response · Authors · 2021-11-19
> **Response to Reviewer Lp8E [Part 1/2: Discussion on GDP]**
>
> We thank the reviewer for the constructive comments and appreciate our theoretical contribution.
>
> **Q1: Some justification of GDP is necessary. What are potential other definitions? Why is this definition the most natural (and technically sound) extension of the categorical definition? Explaining in simple terms would help provide intuition for the reader.**
>
> A1: Thanks for your constructive comment. For fairness metric over continuous attributes, existing fairness metrics rely on the statistical measurement of independence, such as Hirschfeld-Gebelein-Renyi (HGR) maximal correlation coefficient [1] and mutual information [2,3]. The shortcoming of existing metrics is that we can not directly adopt the estimated probability to calculate the mutual information since **the definition of mutual information involved the ratio of probability density function**. Therefore, the estimated mutual information is not robust over the estimated probability density function, especially for the value with low probability density.
>
> Existing methods to estimate mutual information rely on functional representation, i.e., mutual information equals maximization over specific objective function, such as Donsker-Varadhan representation [4]. In other words, existing methods suffers from the **intractable computation** due to usually involved **functional optimization**.
> Subsequently, a neural network is trained to approximate the maximal objective function. A similar neural network approximation trick is also developed for HGR estimation in [5]. Singular value decomposition-based estimation for HGR is also proposed in [1]. Overall, the existing estimation methods either rely on training-needed neural network approximation or singular value decomposition and thus is computation-burden.
>
> In a nutshell, the superiority of our GDP is the robustness over estimated probability density function since the definition only involves **minus and multiply operation**. We also provide the GDP estimation error **convergence proof** and validate the effectiveness of our estimation method in synthetic experiments.
>
> **Natural extension justification** Existing fairness metrics HGR coefficient and mutual information are naturally applicable for continuous and discrete attributes. However, these metrics for discrete attributes is **inconsistent** and **incompatible** with the common-used demographic parity (DP) [6] and difference w.r.t. demographic parity (DDP) [7]. We have shown that our GDP definition is **equivalent** to DP and DDP for binary and categorical sensitive attributes. Therefore, our GDP is a natural extension for categorical attributes. We have added the discussion of the relationship between GDP with DP and DDP on pages 3 and 4.
>
>
> [1] J. Mary, et al. Fairness-aware learning for continuous attributes and treatments. In International Conference on Machine Learning, pp. 4382–4391. PMLR, 2019.
>
> [2] Akshita Jha, et al. Fair representation learning using interpolation enabled disentanglement. arXiv preprint arXiv:2108.00295, 2021.
>
> [3] Elliot Creager, et al. Flexibly fair representation learning by disentanglement. In International conference on machine learning, pp. 1436–1445. PMLR, 2019.
>
> [4] Mohamed Ishmael Belghazi, et al. Mutual information neural estimation. In International Conference on Machine Learning, pp. 531–540. PMLR, 2018.
>
> [5] Vincent Grari, et al. Fairness-aware neural renyi minimization
> for continuous features. In Proceedings of the Twenty-Ninth International Joint Conference on Artificial Intelligence (IJCAI-20)
>
> [6] Michael Feldman, et al. Certifying and removing disparate impact. In proceedings of the 21th ACM SIGKDD
> international conference on knowledge discovery and data mining, pp. 259–268, 2015.
>
> [7] Cho, Jaewoong, et al. "A fair classifier using kernel density estimation." NeurIPS 2020.

---

> ### Author Response · Authors · 2021-11-19
> **(Continued) Response to Reviewer Lp8E [Part 2/2: Discussions on age attributes, other related metrics, and more experiments]**
>
> **Q2: Section 1: Age is not a particularly successful example of a continuous attribute. A better example might be income differences (although it is not always legally protected, it is definitely of interest to decision-makers).**
>
> A2: Thanks for your constructive comment. We think that **age is a continuous attribute** even if the age attribute value is integral and therefore is not continuous. We also agree that income is a better example for the continuous attribute even if the income value is not strictly continuous since there is the smallest unit for income value. **The biggest difference between continuous and discrete sensitive attribute falls in the existence of order information [1] for attribute values.** For example, $50 k/year$ income is lower than $60 k/year$ income. For race or gender attributes, we cannot compare different attribute values since there is no order information. Therefore, we argue that income attribute is continuous, and race or gender attribute is discrete. In other words, the pair of continuous attribute values similarity is usually **diverse** for different pairs. Differently, the pair discrete attributes similarity is usually the same. **The challenge of fairness over continuous attribute is that how we can tackle the order information in continuous attributes.** Based on these observations, we think that age is a continuous attribute. We have added the discussion on page 7.
>
> [1] Mary, et al. Fairness-aware learning for continuous attributes and treatments. In International Conference on Machine Learning, pp. 4382–4391. PMLR, 2019.
>
> **Q3: It is important to include a discussion on how other metrics can be similarly extended to capture continuous attributes and add a proof sketch or intuition after each theorem.**
>
> A3: Thanks for your constructive comment. We have added the discussion on the equal odds extension to continuous attributes and added a proof sketch and intuition after each theorem. For example, we have added the proof sketch on GDP estimation error bounds. The proof sketch is to **separately bound the error for local average estimation and sensitive attribute probability density estimation**. Finally, we combine these two estimations to obtain the estimation error for GDP. We also provide the intuition for the connection between GDP and adversary utility. **Intuitively, a fair classifier aims to minimize the adversary utility to make prediction fair, and thus induces lower GDP metric.** Therefore, it is possible that there exists an inherent connection between GDP and adversary utility. Please see pages 5 and 6 for more details.
>
>
> **Q4: Section 6.1: why were these parameters chosen? Have robustness checks been performed and what were the results?**
>
> A4: Thanks for your constructive comment. Due to limited space, we only choose one specific parameter to validate the effectiveness of our proposed estimation methods. For validating the estimation error robustness over different parameters, we have also added the robustness estimation error over different mean and variance parameters in Appendix G.3. It is seen that kernel-based estimation methods consistently achieve lower estimation error than histogram-based method, and kernel-based estimation method is **not sensitive** to kernel function choice.

---

> ### Author Response · Authors · 2021-11-23
> **Appreciate further comments**
>
> Dear Reviewer Lp8E,
>
> We have responded to your follow-up questions. We are looking forward to your re-assessment of our work and are happy to have further discussions to address your concerns.
>
> Sincerely, Authors

---

> ### Comment · Reviewer_Lp8E · 2021-11-30
> **thank you for your response**
>
> Thank you for your detailed response to my comments. My positive impression of the paper has only been reinforced. At least for now, I will maintain my current score.

---

### Official Review · Reviewer_s7Ek · 2021-11-02

**Correctness:** 3
**Technical Novelty And Significance:** 3
**Empirical Novelty And Significance:** 3
**Recommendation:** 6
**Confidence:** 4

**Main Review:**

This paper proposes the generalized demographic parity (GDP). It aims to generalize the existing definition of demographic parity (DP) to incorporate continous sensitive attribute while preserving tractable computation. More specifically, GDP is defined as the weighted total variation distance between local prediction average and global prediction average, with the weights being the probability density function (PDF) of sensitive attribute. Based on the definition, histogram estimation and KDE are applied to estimate the distribution of sensitive attribute. In addition, the authors shows that, under certain assumptions, (1) GDP is equivalent to DP; and (2) GDP is the lower bound of adversarial utility by Madras et al. 2018. Experiments on several real-world datasets across different settings demonstrate the effectiveness of the proposed method against baseline methods. However, I have several concerns about this manuscript as shown below.

Concerns
- The authors claim that GDP is tractable. I wonder how it is tractable if we do not know the joint distribution of prediction and sensitive attribute (as claimed by the first few sentences in Section 4).
- The proposed method uses histogram/KDE to estimate the distribution of sensitive attribute. Why cannot we use the same techniques (i.e., histogram/KDE) to estimate the distributions needed for calculating mutual information (e.g., joint distribution of prediction and sensitive attribute,  conditional distribution of prediction given sensitive attribute, or conditional distribution of sensitive attribute given predictions)?
- The experiments are not convincing. (1) The authors established the relationship between GDP and adversarial debiasing by Madras et al. 2018. Why is there no experiments on comparing the performance between these two methods? (2) The paper barely discusses the work Louppe et al. 2017 but includes it as the only debiasing method for comparison. I wonder the justification behind the choice of baseline methods. (3) Mary et al. 2019 also works on debiasing the continuous sensitive attribute, yet there is no comparison. I believe certain justification is needed as well.
- Some related works are missing (please see below). And I believe some of them should be included as baseline methods as well.

References

[Related to fairness with histogram model]

* Kamishima, T., Akaho, S., Asoh, H., & Sakuma, J. (2012, September). Enhancement of the Neutrality in Recommendation. In Decisions@ RecSys (pp. 8-14).
* Kamishima, T., Akaho, S., Asoh, H., & Sakuma, J. (2013, September). Efficiency Improvement of Neutrality-Enhanced Recommendation. In Decisions@ RecSys (pp. 1-8).

[Related to fairness with KDE]

* Cho, J., Hwang, G., & Suh, C. (2020). A fair classifier using kernel density estimation. Advances in Neural Information Processing Systems, 33, 15088-15099.

[Related to fairness with mutual information]

* Cho, J., Hwang, G., & Suh, C. (2020, June). A fair classifier using mutual information. In 2020 IEEE International Symposium on Information Theory (ISIT) (pp. 2521-2526). IEEE.
* Roh, Y., Lee, K., Whang, S., & Suh, C. (2020, November). Fr-train: A mutual information-based approach to fair and robust training. In International Conference on Machine Learning (pp. 8147-8157). PMLR.
* Lowy, A., Pavan, R., Baharlouei, S., Razaviyayn, M., & Beirami, A. (2021). FERMI: Fair Empirical Risk Minimization via Exponential R\'enyi Mutual Information. arXiv preprint arXiv:2102.12586.

**Summary Of The Paper:**

Please see main review.

**Summary Of The Review:**

Overall, this paper is a bit unclear in some parts and lacks a thorough literature review. The experiments are not convincing enough as well.

---

> ### Author Response · Authors · 2021-11-19
> **Response to Reviewer s7Ek [Part 1/2: tractability of GDP]**
>
> Thanks for your constructive reviews. It seems that you have a negative attitude toward our work. To address your concerns, we give the following point-to-point responses. We believe that if you read it carefully, you will change your attitude.
>
> **Q1: The authors claim that GDP is tractable. I wonder how it is tractable if we do not know the joint distribution of prediction and sensitive attribute (as claimed by the first few sentences in Section 4).**
>
> A1: Thanks for your constructive comment. We claim that GDP is tractable since GDP can be **easily** estimated via the joint distribution of prediction and sensitive attribute estimation with **theoretical guarantee**. Specifically, we proposed histogram-based and kernel-based estimation methods to obtain GDP estimation. We further provably and numerically demonstrate the convergence of estimation error for these two methods.
>
> **Comparison with mutual information**: The mutual information can be also adopted to measure the fairness metric over continuous sensitive attributes. However, the mutual information definition involves **the ratio of probability density function**. The estimated probability density function is not robust over the probability density function, especially for the value with low probability density. In other words, **directly adopting the estimated probability density function to calculate mutual information may not be reliable.** Different from mutual information, the definition of GDP only involves **minus and multiply operations**, and thus GDP calculation is robust to the perturbation of probability density function.
>
> **Fairness metric estimation:** Existing methods to estimate mutual information rely on functional representation, i.e., mutual information equals maximization over specific objective function, such as Donsker-Varadhan representation [1]. Subsequently, **a neural network is trained to approximate the maximal objective function.** The shortcoming of functional representation is three-fold. First, the estimated mutual information is **always lower than** the true mutual information since the neural network may not achieve the maximal objection function. There is **not any convergence guarantee** for the biased mutual information estimation. Second, the mutual information estimation is **computation-consuming** since neural network training is needed for each fairness metric evaluation.
>
> Therefore, we claim that GDP is tractably attributed to the robustness over probability density function distribution and thus avoid to involve in the complex estimation method. In the revised version, we have added the discussion on the tractability of GDP on the footnote of page 1.
>
> [1] Mohamed Ishmael Belghazi, et al. Mutual information neural estimation. In International Conference on Machine Learning, pp. 531–540. PMLR, 2018.

---

> ### Author Response · Authors · 2021-11-19
> **(Continued) Response to Reviewer s7Ek [Part 2/2: more discussions and experiments]**
>
> **Q2: The proposed method uses histogram/KDE to estimate the distribution of sensitive attributes. Why cannot we use the same techniques to estimate the distributions needed for calculating mutual information?**
>
> A2: Thanks for your constructive comment. As we mentioned in A1, **directly adopting estimated probability density function to estimated mutual information is not reliable** since the definition of mutual information involves the ratio of probability density functions, and the estimated probability density function is not accurate, especially for the low probability density value.
>
> The superiority of GDP falls in the **tractability and reliability** of GDP. The robustness of GDP over the perturbation of probability density function eases the GDP estimation via directly adopting probability density function estimation methods. The reliability of GDP represents the estimation error **convergence guarantee** w.r.t. data size. For the existing mutual information estimation, such as MINE [1], there is not any convergence guarantee for neural network approximation. In the revised manuscript, we have added the justification on page 1.
>
> [1] Mohamed Ishmael Belghazi, et al. Mutual information neural estimation. In International Conference on Machine Learning, pp. 531–540. PMLR, 2018.
>
> **Q3: Experiments: (1)The experiments are not convincing. There are no experiments on comparing the performance between GDP and adversarial debiasing by Madras et al. 2018. The paper only compares adversarial debiasing by Louppe et al. 2017.  (2) More baselines are beneficial, such as Mary et al. 2019.**
>
> A3: We are sorry for the confusion on adversarial debiasing. We only adopt the adversary utility function concept proposed in adversarial debiasing [2]. The adversarial debiasing [2] is actually proposed for **transferable representation learning** and the model adopts **encoder-decoder structure**. In other words, the target task and model in adversarial debiasing [2] are both different from that in our work. As for adversarial debiasing [3], we actually adopt **the same model structure as adversarial debiasing [3]**, including classifier and adversary, where the input of adversary is the output of the classifier. Therefore, we compare the performance with adversarial debiasing [3].
>
> In addition, we have added the baseline [4], which derives an upper bound of Hirschfeld-Gebelein-Renyi (HGR) as a regularizer. It is seen that our GDP regularizer can still achieve the best tradeoff performance with the best stability. The key reason is that the upper bound of Hirschfeld-Gebelein-Renyi (HGR) still involves the ratio of probability and thus is not robust over perturbation of probability density function. Please see pages 6 and 7.
>
> [2] Madras, et al. Learning adversarially fair and
> transferable representations. In International Conference on Machine Learning, pp. 3384–3393.
> PMLR, 2018.
>
> [3] Louppe, et al. Learning to pivot with adversarial networks. In
> Proceedings of the 31st International Conference on Neural Information Processing Systems, pp.
> 982–991, 2017.
>
> [4] Mary, et al. Fairness-aware learning for continuous attributes and treatments. In International Conference on Machine Learning, pp. 4382–4391. PMLR, 2019.
>
>
> **Q4: Some related works are missing.**
>
> A4: Thanks for your constructive comment. We have added the discussion on these related works on page 1.

---

> ### Author Response · Authors · 2021-11-21
> **Appreciate further comments**
>
> Dear Reviewer s7Ek,
>
> We really appreciate your previous constructive comments and have addressed your concerns one by one. As the discussion period is approaching its end, we would really appreciate it if you could kindly let us know there are any further questions. We will be more than happy to address them fully.
>
> Yours Sincerely,
>
> Authors

---

> > ### Comment · Reviewer_s7Ek · 2021-11-22
> > **My response**
> >
> > Thank you for addressing my concerns. Most of my concerns are addressed. Here are some of my further concerns and suggestions.
> >
> > 1. When you talk about Mary et al. in introduction, you mentioned it relies on the complex singular value decomposition. Why is it more complex than using KDE for pdf estimation?
> >
> > 2. I think it would be better to highlight the differences between Madras et al. and adversarial debiasing in the paper (like your responses to me). Simply saying `'similar to xxx, ...`' could still be somewhat misleading.
> >
> > 3. If the space allows, I would suggest move some footnotes into the main body, especially when you talk about intractability and unreliability of mutual information estimation.
> >
> > 4. Please carefully proofread the paper, there are some typos and grammatical errors.

---

> > > ### Author Response · Authors · 2021-11-22
> > > **Response to Reviewer s7Ek**
> > >
> > > Thanks for your constructive comments.
> > >
> > > Q1: Why is singular value decomposition more complex than using KDE for pdf estimation?
> > >
> > > **A1:** In the Introduction part of Section 1, we say that SVD is computationally complex since the computation complexity of HGR is higher than GDP due to the involved SVD. In Theorem 2.2 of work [1], the authors mention that HGR coefficient $HGR(U,V)$ measuring independence of random variables $U$ and $V$ satisfies $HGR(S,\hat{Y})=\sigma_2(Q)$, where $\sigma_2$ is the second-largest singular value of a matrix and distribution ratio matrix $Q(u,v)=\frac{P_{U,V}(u,v)}{P_U(u)P_V(v)}$. Therefore, to calculate HGR, we need to estimate the two-dimensional probability density function to obtain the distribution ratio matrix $Q$ and then calculate the second-largest singular value.
> > >
> > > For a fair computation complexity comparison, we assume there are $M$ probing points in continuous sensitive attributes (we estimate $M$ pdf value for each dimension) and $N$ data points. The computation complexity of **two-dimensional KDE estimation** is $O(M^2N)$, and that of SVD for $M\times M$ matrix $Q$ is $O(M^3)$. Therefore, the computation complexity for HGR is $O(M^2(M+N))$. As for **histogram or kernel-based estimation**, we only need to estimate the one-dimensional sensitive attribute pdf, local and global average with the computation complexity $O(MN)$, $O(MN)$, and $O(N)$, respectively. Therefore, the computation complexity for GDP is $O(MN)$.
> > >
> > > In a nutshell, SVD-based HGR calculation has higher computation complexity than GDP calculation due to the involved SVD. Please see the last part of Section 4 for more details.
> > >
> > > [1] Mary, et al. Fairness-aware learning for continuous attributes and treatments. In International Conference on Machine Learning, pp. 4382–4391. PMLR, 2019.
> > >
> > > Q2: I think it would be better to highlight the differences between Madras et al. and adversarial debiasing in the paper (like your responses to me). Simply saying 'similar to xxx, ...' could still be somewhat misleading.
> > >
> > > **A2:** We have added more description on the model and objective function of adversary debasing in Section 5. Additionally, we also illustrated that we only adopt utility function in (Madras et al. 2018) to connect GDP and adversarial debiasing (Louppe et al., 2017).
> > >
> > >
> > > Q3: If the space allows, I would suggest moving some footnotes into the main body, especially when you talk about intractability and unreliability of mutual information estimation.
> > >
> > > **A3:** Thanks for your constructive comments. We have moved the footnote on the intractability and unreliability of mutual information estimation in the Introduction Section. Please see page 1.
> > >
> > > Q4: Please carefully proofread the paper, there are some typos and grammatical errors.
> > >
> > > **A4:** We have carefully proofread this paper and revised several typos and grammatical errors.

---

> > > ### Author Response · Authors · 2021-11-22
> > > **Thanks very much for your positive feedback**
> > >
> > > Dear Reviewer s7Ek:
> > >
> > > Thanks for your positive re-assessment of our work. We enjoyed the fruitful discussion with you. Please do let us know if there is anything that you believe we can do to improve this work!
> > >
> > > Best, Authors

---

### Official Review · Reviewer_zy7G · 2021-11-03

**Correctness:** 4
**Technical Novelty And Significance:** 3
**Empirical Novelty And Significance:** 3
**Recommendation:** 6
**Confidence:** 3

**Main Review:**

Strength:

1. The GDP is a generic fairness metric for both continuous and discrete protected attributes.
2. The histogram and kernel estimations are computationally feasible. The error and complexity analysis of the two estimations show improvement.
3. The connection of GDP and adversarial debiasing is built.
4. The comprehensive experiments on multiple datasets show the efficiency and effectiveness of the kernel estimation of GDP.

Weakness:

1. There is a gap between the estimation of GDP and adversarial debiasing. Vanilla GDP relies on an unknown distribution which prevents its application. If the estimation is applied, it is unclear the relationship between $\hat{GDP}$ and the adversary utility.
2. It is unclear how to calculate the underlying GDP in the experiments, or it is just an estimation of GDP. Please clarify.


**Summary Of The Paper:**

This paper proposes a generalized demographic parity for group fairness which is computationally feasible for both continuous and discrete protected attributes. Two estimations, histogram and kernel, are proposed for efficient estimation and the kernel estimation has faster estimation error governance. The connection between the GDP regularization and adversarial debiasing is built. The experiments on syntehtic/tabular/graph datasets show the effectiveness and efficiency of the GDP kernel estimation.

**Summary Of The Review:**

This paper provides a generic fairness metric for both continuous and discrete protected attributes in various tasks. The kernel estimation has good error and complexity properties. The experimental comparison is comprehensive and well designed.

The connection between GDP and binary demographic parity is built, but I wonder about the relationship between GDP and discrete protected attributes with multiple values, e.g., intersectional fairness. The extension from binary to multiple to continuous would be more smooth.

---

> ### Author Response · Authors · 2021-11-19
> **Response to Reviewer zy7G**
>
> We thank the reviewer for the constructive comments and appreciate the reviewer for the recognition of the effectiveness and efficiency of our work.
>
> **Q1: There is a gap between the estimation of GDP and adversarial debiasing. Vanilla GDP relies on an unknown distribution which prevents its application. If the estimation is applied, it is unclear the relationship between $\hat{GDP}$ and the adversary utility.**
>
> A1: Thanks for your insightful comment. We extend Theorem 3.4 and show that the connection between GDP and adversarial utility can **not only hold for unknown underlying data distribution but also empirical data distribution**. Note that the data distribution is the same for GDP and adversarial utility (either underlying data distribution or empirical data distribution), it is very easy to extend the established connection replacing underlying data distribution as empirical data distribution in the expectation operation (See Appendix E for more detail). We have added the discussion that the established connection between GDP and adversarial utility even holds for empirical data distribution on page 6.
>
> **Q2: It is unclear how to calculate the underlying GDP in the experiments, or it is just an estimation of GDP. Please clarify.**
>
> A2: Thanks for your constructive comment. We adopt estimated GDP via kernel function to quantify the fairness metric in the experiments on real-world datasets. It is almost impossible to obtain the underlying GDP or evaluate GDP estimation on real-world datasets since there is no ground truth of underlying data distribution. Alternatively, **we validate the effectiveness of our proposed GDP estimation methods on two synthetic datasets.** It is seen that kernel GDP estimation has a lower GDP estimation error. We have illustrated GDP estimation on the footnote of page 6.
>
> Additionally, **the synthetic experiments show that the GDP estimation error over the true GDP is less than $2%$ when data size is larger than $2000$.**  In Appendix F, we provide the statistical information on real-world datasets and 4 dataset size largely exceed 2000, and only the size of Crimes dataset is 1994. Therefore, we believe that kernel GDP estimation is **sufficiently reliable** to measure the fairness metric on real-world datasets. It is also worthwhile to mention that, even for well-known demographic parity (DP), the underlying DP is still unavailable and the estimated DP over empirical data distribution is adopted in real-world datasets. Similarly, for the sufficiently large datasets, the estimated fairness metric is sufficiently reliable to measure the fairness metric.
>
>
> **Q3: The relationship between GDP and discrete protected attributes with multiple values, e.g., intersectional fairness is missing. The extension from binary to multiple to continuous would be more smooth.**
>
> A3: Thanks for your constructive comment. We have added the discussion on the relationship between GDP and categorical protected attributes (with multiple values). Previous work [1] introduces **difference w.r.t. demographic parity (DDP)** to measure the fairness metric over ** only discrete protected attributes with multiple or binary values**, where $DDP$ is defined as the summation of the difference between local and global prediction average $DDP=\sum_{z\in\mathcal{Z}}|P(\tilde{Y}=1|z)-P(\tilde{Y}=1)|$, $\mathcal{Z}$ is defined as sensitive attribute value set, $Z$ and $\tilde{Y}$ represent sensitive attribute and model prediction.
> It is easy to show that our GDP definition exactly equals DDP for discrete protected attributes with multiple values. Therefore, the proposed GDP definition is a general fairness metric applicable for binary or categorical protected attributes and continuous protected attributes. We have added the discussion on the relation between GDP and DDP on page 4 of the revised version.
>
>
> [1] Cho, Jaewoong, Gyeongjo Hwang, and Changho Suh. "A fair classifier using kernel density estimation." NeurIPS 2020.

---

> ### Author Response · Authors · 2021-11-23
> **Appreciate further comments**
>
> Dear Reviewer zy7G,
>
> Thank you for your valuable suggestions again. We have responded to your initial comments. We are looking forward to your feedback and are glad to answer your further questions.
>
> Thanks, Authors

---

### Author Response · Authors · 2021-11-20
**To All Reviewers**

Dear Reviewers,

Thanks very much for your constructive suggestions and for appreciating the soundness of our work. We respectfully accept your valuable suggestions and added more baselines. We add **HGR performance trade-off results in Section 6 and estimation error robustness experiments in Appendix G.3**. Thank you for your careful reading in advance.

To address your major concerns about this work, we try our best to provide several improvements, including

- Extend GDP and adversarial debiasing not only for underlying data distribution but also for **empirical data distribution**.

- Justify why Lipschitz continuous assumption is **not a strong assumption and the significance of Theorem 3**.

- Add baselines HGR [1] to show the effectiveness of our GDP in the experiments and estimation error robustness experiments over distribution parameters.

- Add the discussion on the relationship between GDP and difference w.r.t. demographic parity (DDP) for **categorical sensitive attributes**.

- Clarify the **superiority of GDP** compared with other metrics, such as mutual information: **the robustness over estimated probability density function**, **theoretical guarantee for estimation error** and **lower computation complexity**.

- Discuss the critical difference between continuous and discrete sensitive attributes to justify the continuous ``age" attribute.

- Add more details on the experimental setting.

Thank you again for your valuable comments and suggestion. We are looking forward to your feedback and are happy to answer your follow-up questions.

[1] J. Mary, et al. Fairness-aware learning for continuous attributes and treatments. In International Conference on Machine Learning, pp. 4382–4391. PMLR, 2019.

---

### Author Response · Authors · 2021-11-21
**Appreciating for your previous efforts and further comments**

Dear AC and Reviewers,

We genuinely thank AC and all the reviewers for your time and constructive comments! Hope our previous responses have addressed your concerns.

To address your major concerns about this work, we try our best to provide several improvements, including

- Extend GDP and adversarial debiasing not only for underlying data distribution but also for empirical data distribution.

- Justify why Lipschitz continuous assumption is not a strong assumption and the significance of Theorem 3.

- Add baselines HGR [1] to show the effectiveness of our GDP in the experiments and estimation error robustness experiments over distribution parameters.

- Add the discussion on the relationship between GDP and difference w.r.t. demographic parity (DDP) for categorical sensitive attributes.

- Clarify the superiority of GDP compared with other metrics, such as mutual information: the robustness over estimated probability density function, theoretical guarantee for estimation error, and lower computation complexity.

- Discuss the critical difference between continuous and discrete sensitive attributes to justify the continuous ``age" attribute.

- Add more details on the experimental setting.

[1] J. Mary, et al. Fairness-aware learning for continuous attributes and treatments. In International Conference on Machine Learning, pp. 4382–4391. PMLR, 2019.

As the discussion period is approaching its end, we would really appreciate it if you could kindly let us know there are any further questions. We will be more than happy to address them fully.

Yours Sincerely,

Authors

---

### Decision · Program_Chairs · 2022-01-20

**Decision:**

Accept (Poster)

**Comment:**

This paper offers an alternative formulation of demographic parity, named GDP, which makes it amenable to easier computation when the sensitive attribute is continuous. Analytically, the paper relates GDP to other notions, offers ways to estimate GDP from data, and establishes the convergence of these estimators. Experimentally, the paper adds the estimated GDP as a learning regularizer and establishes the accuracy-fairness tradeoff that results by using this method versus others.

The need to handle continuous sensitive attributes is well-motivated since they are ubiquitous. The direction of the paper is thus very pertinent. The experimental exploration of the paper is also strong, though reviewers initially raised questions of clarity of the relationship of GDP with adversarial debiasing. These are mostly addressed by the authors. One weakness of the paper that largely remains is whether the paper offers new conceptual insights. Indeed, demographic parity is simply a notion of independence between an algorithm’s output and sensitive attributes. Other independence metrics are dismissed in the paper as unreliable to compute. However, one reviewer correctly raises the concern that *under similar regularity conditions* to the ones establishing the convergence of the kernel GDP estimator, it is also possible to establish convergence of other independence metrics, that would equally capture demographic parity. Another reviewer also points out that such convergence would follow using standard non-parametric statistics techniques. Smoothed estimators of mutual-information are indeed available in the literature, with convergence guarantees even in the high-dimensional regime. The authors do not satisfactorily address this, casting doubt on the overall significance of the contribution.

That said, given the strong motivation behind the paper and the overall promise of the methodology, it may be worth sharing with the community. The authors are urged to address the above. Additionally, they are urged to be transparent about what the theory offers and what it doesn’t. For instance, the convergence results of GDP only tell us that we can use these estimators to audit the fairness of existing models. In other words, although the paper is touted as showing that GDP can be successfully used for learning, the evidence there is purely empirical: there is no learning guarantee simultaneously on the accuracy and fairness of GDP-penalized risk minimization.